# Full-Rank Unsupervised Node Embeddings for Directed Graphs via Message Aggregation

**Ciwan Ceylan**                                                              *ciwan@kth.se*
*Division of Robotics, Perception and Learning*
*KTH Royal Institute of Technology*

**Kambiz Ghoorchian**                                           *kambiz.ghoorchian@seb.com*
*SEB Group*

**Danica Kragic**                                                              *dani@kth.se*
*Division of Robotics, Perception and Learning*
*KTH Royal Institute of Technology*

**Reviewed on OpenReview:** *https://openreview.net/forum?id=3ECbEZg2If*

## Abstract

Linear message-passing models have emerged as compelling alternatives to non-linear graph neural networks for unsupervised node embedding learning, due to their scalability and competitive performance on downstream tasks. However, we identify a fundamental flaw in recently proposed linear models that combine embedding aggregation with concatenation during each message-passing iteration: rank deficiency. A rank-deficient embedding matrix contains column vectors which take arbitrary values, leading to ill-conditioning that degrades downstream task accuracy, particularly in unsupervised tasks such as graph alignment. We deduce that repeated embedding aggregation and concatenation introduces linearly dependent features, causing rank deficiency. To address this, we propose ACC (Aggregate, Compress, Concatenate), a novel model that avoids redundant feature computation by applying aggregation to the messages from the previous iteration, rather than the embeddings. Consequently, ACC generates full-rank embeddings, significantly improving graph alignment accuracy from 10% to 60% compared to rank-deficient embeddings, while also being faster to compute. Additionally, ACC employs directed message-passing and achieves node classification accuracies comparable to state-of-the-art self-supervised graph neural networks on directed graph benchmarks, while also being over 70 times faster on graphs with over 1 million edges.

## 1 Introduction

Node embeddings, which represent nodes in a graph as vectors, have proven highly effective for various graph-related tasks, including node classification (Veličković et al., 2018; Rossi et al., 2023), node clustering (Henderson et al., 2012; Donnat et al., 2018), and graph alignment (Heimann et al., 2018; Skitsas et al., 2023). Consequently, there has been substantial research focused on developing algorithms to compute these embeddings, resulting in a wide array of models (Kipf & Welling, 2017; Wu et al., 2019; 2021; Rossi et al., 2023).

Given the scarcity of labelled data in real-world graphs (Veličković et al., 2019), our work focuses on unsupervised embedding models. Unlike supervised models, which target specific downstream tasks, unsupervised models aim to extract and compress the information inherent in the graph structure into individual embedding vectors. The dominant approach for this extraction is *message-passing* (Gilmer et al., 2017).

In a message-passing algorithm, each node is initially assigned a feature vector, serving as its initial embedding. These embeddings are iteratively refined by incorporating information from increasingly larger

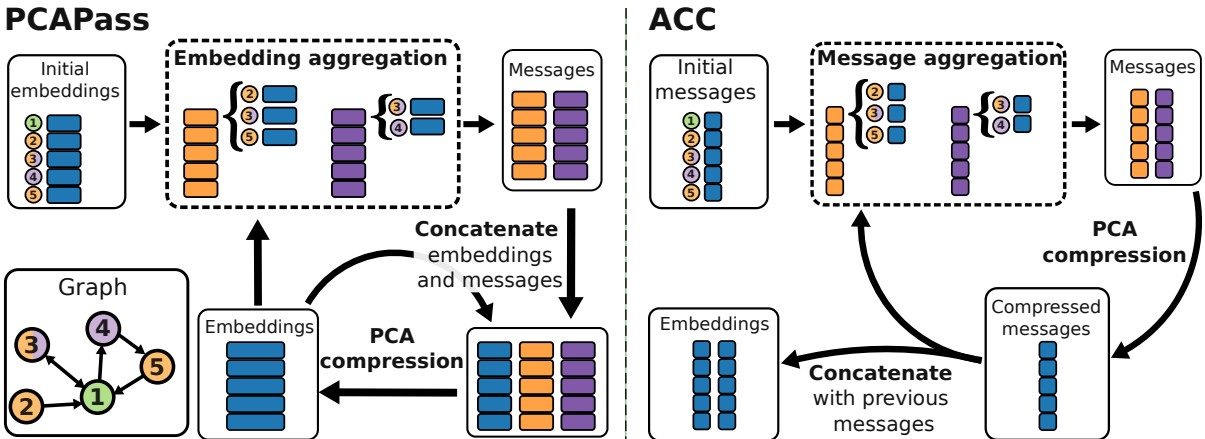

Figure 1: Overview of directed message-passing in PCAPass (Sadowski et al., 2022) and our model, ACC. Each vertically stacked rectangle represents a matrix row corresponding to a node in the graph (bottom-left). The representations are colour-coded: blue for node embeddings, and orange and purple for messages. Orange denotes aggregation following edge directions, while purple indicates reverse-direction aggregation. Node 1, highlighted in green in the graph, is used to illustrate these aggregations. In PCAPass, messages are concatenated with the embeddings from the previous iteration, creating a feedback loop that leads to feature duplication and rank deficiency over multiple iterations. ACC avoids this issue through message aggregation, where only the message matrices are propagated between iterations. The embedding matrix is constructed by concatenating the messages outside the feedback loop.

neighbourhoods. Conceptually, each iteration consists of two steps: the aggregation step, where information from a node's neighbourhood is summarized into a message, and the update step, where these messages are integrated into the existing embeddings.

When the aggregation and update steps consist of parameterized and non-linear functions, the message-passing model is referred to as a Graph Neural Network (GNN) (Wu et al., 2021). While the parameterization and non-linearity of GNNs make them highly expressive, these features also limit their scalability. Training GNNs requires non-convex optimization, typically through gradient descent, to minimize a self-supervised loss function. This training process involves hundreds or even thousands of training epochs, each consisting of multiple message-passing iterations (Zhang et al., 2021; Thakoor et al., 2022; Hou et al., 2022; 2023).

In their seminal work, Wu et al. (2019) addressed the scalability challenge of GNNs by introducing a linear message-passing model called SGCN. They demonstrated that SGCN could achieve accuracies comparable to GNNs on popular node classification benchmarks while being significantly more scalable, requiring only a single execution of the message-passing procedure. However, despite these advancements, SGCN encounters a common issue in message-passing models: over-smoothing (Li et al., 2018; Chen et al., 2020a), where the repeated summation of embeddings across iterations leads to a gradual loss of information.

Computing *multi-scale* embeddings, i.e., concatenating vectors that capture information from various neighbourhood sizes, is a common method for mitigating information loss in unsupervised node embedding models (Tang et al., 2015; Cao et al., 2015; Perozzi et al., 2017; Donnat et al., 2018; Rozemberczki et al., 2021). A recent example of this approach is PCAPass (Sadowski et al., 2022). As shown on the left in Figure 1, PCAPass aggregates node embeddings in each message-passing iteration to create new features, called messages, which are then combined with the existing embeddings via horizontal concatenation. To prevent the embedding dimensionality from doubling with each iteration, the concatenated matrix is compressed using PCA.

While the concatenation approach addresses over-smoothing, it introduces a new issue: rank deficiency (Hansen, 1998, Ch. 1). A rank-deficient matrix is characterized by a cluster of singular values close to zero, each corresponding to an arbitrary and non-informative feature column. This phenomenon affects the PCAPass embedding matrix and degrades the quality of the embeddings. Unsupervised downstream tasks, which rely on distances between embeddings, are particularly sensitive to this issue, raising concerns since such tasks are well-suited for unsupervised node embedding models.

We identify that the repeated embedding aggregation and concatenation in PCAPass leads to the creation of linearly dependent features, which results in rank deficiency. To address this, we propose ACC[1] (Aggregate, Compress, Concatenate), a linear message-passing model designed to prevent rank deficiency while maintaining scalability by generating embeddings in a single forward pass. As shown in Figure 1, ACC applies aggregation and PCA compression to the *message matrices* from the previous iteration, rather than the embeddings, and constructs the embeddings via concatenation separately. This *message aggregation* approach breaks the feedback loop present in PCAPass, and thereby avoids computing the redundant features that cause rank deficiency.

In support of ACC, we demonstrate the rank deficiency issue present in PCAPass, focusing particularly on its negative impact on unsupervised embedding-based graph alignment (Heimann et al., 2018). Although it is technically possible to address the rank deficiency in PCAPass through singular value thresholding, this approach is not only difficult due to numerical inaccuracies but also inefficient, as computation is used to generate the redundant features in the first place. Consequently, ACC consistently achieves higher graph alignment accuracies and computes embeddings faster than PCAPass.

We also demonstrate ACC's effectiveness in learning node embeddings on directed graphs, an underexplored challenge for unsupervised message-passing models that has only recently been addressed in the supervised learning context (Rossi et al., 2023). Using standard directed graph node classification benchmarks without hyperparameter tuning, we show that ACC achieves accuracies comparable to state-of-the-art self-supervised GNNs. Notably, ACC is significantly faster, at least 70 times faster on the Arxiv Year dataset (Lim et al., 2021) with GPU computations and 270 times faster on Snap Patents (Leskovec et al., 2005) with CPU.

## 2 Background: Node embeddings via message-passing and embedding aggregation

The message-passing framework (Gilmer et al., 2017) forms the basis for a wide range of graph models (Kipf & Welling, 2017; Hamilton et al., 2017; Wu et al., 2019), including our model, ACC. In this section, we introduce the principles and mathematical notation for node embedding learning through message-passing, followed by a description of the embedding aggregation and concatenation approach used by Sadowski et al. (2022) for PCAPass. Additionally, we outline directed message-passing for PCAPass, following Rossi et al. (2023).

Node embedding message-passing models take as input a graph $\mathcal{G} = (\mathbb{N}, \mathbb{M})$ and a matrix $\boldsymbol{X} \in \mathbb{R}^{n \times d}$, and output an embedding matrix $\boldsymbol{Z} \in \mathbb{R}^{n \times p}$. The graph $\mathcal{G}$ consists of $n$ nodes $\mathbb{N}$, with $n = |\mathbb{N}|$, connected by $m$ edges $\mathbb{M}$, where $m = |\mathbb{M}|$. The matrix $\boldsymbol{X}$ contains initial feature vectors of length $d$ for each node, and the embedding matrix $\boldsymbol{Z}$ holds the resulting $p$-dimensional node embeddings. In this work, we assume a transductive setting where $\mathcal{G}$ is fully observed during the computation of $\boldsymbol{Z}$, which is common in unsupervised node embedding models (Zhang et al., 2021; Thakoor et al., 2022; Hou et al., 2022; Sadowski et al., 2022).

The goal of message-passing is to gather graph structure information into the embeddings of each node. This is achieved by iteratively updating each node embedding by incorporating information from the nodes' respective neighbourhood, resulting in a sequence of progressively refined embedding matrices. We denote the embedding matrix after $k$ message-passing iterations as $\boldsymbol{H}^{(k)} \in \mathbb{R}^{n \times p_k}$, where the embedding dimensionality $p_k$ may vary across iterations. The final embeddings are obtained after $K$ iterations, represented as $\boldsymbol{Z} = \boldsymbol{H}^{(K)}$.

Each message-passing iteration consists of two key steps: *aggregation* and *update*. In the aggregation step, each node collects information from its immediate neighbours, aggregating these inputs into a new vector called a message. We denote the matrix of all messages at iteration $k$ as $\boldsymbol{M}^{(k)}$. In the update step, each node integrates the new information by combining its received message with its current embeddings. This iterative process allows information to propagate through the graph, enabling each node to gather data from increasingly distant nodes, effectively expanding its receptive field with each iteration.

We now describe the aggregation and update operations used by PCAPass in detail. Let $\boldsymbol{A} \in \mathbb{R}^{n \times n}$ represent the graph's adjacency matrix, where the element $A_{i,j}$ indicates the presence of an edge from node $j$ to node $i$.

---

[1]ACC model code: https://github.com/ciwanceylan/acc-mp.
Experiments code: https://github.com/ciwanceylan/acc-experiments-tmlr2024.

The out-degree and in-degree of node $i$ are denoted as $\deg_{\mathtt{0}}(i)$ and $\deg_{\mathtt{I}}(i)$, respectively:

$$A_{i,j} = \begin{cases} 1 & \text{if } (j,i) \in \mathbb{M}, \\ 0 & \text{otherwise,} \end{cases} \qquad \deg_{\mathtt{0}}(i) = \sum_{k=1}^{n} A_{k,i}, \qquad \deg_{\mathtt{I}}(i) = \sum_{k=1}^{n} A_{i,k}. \qquad (1)$$

Additionally, let $\boldsymbol{D}_{\mathtt{0}}$ be a diagonal matrix containing the out-degrees, with $D_{\mathtt{0}i,i} = \deg_{\mathtt{0}}(i)$, and similarly, let $\boldsymbol{D}_{\mathtt{I}}$ be a diagonal matrix of in-degrees. The matrices $\boldsymbol{D}_{\mathtt{0}}^{-1}$ and $\boldsymbol{D}_{\mathtt{I}}^{-1}$ represent their respective inverses, with elements corresponding to nodes with zero out-degree or in-degree set to 0.

For undirected graphs, the adjacency matrix is symmetric, $\boldsymbol{A} = \boldsymbol{A}^{\mathsf{T}}$, and each node has a single degree, so $\boldsymbol{D} = \boldsymbol{D}_{\mathtt{0}} = \boldsymbol{D}_{\mathtt{I}}$. The normalized adjacency matrix is then defined as $\boldsymbol{A}_{\mathtt{N}} = \boldsymbol{D}^{-1}\boldsymbol{A}$. Using the above definitions, the embedding aggregation step used by PCAPass can be formulated as $\boldsymbol{M}^{(k)} = \boldsymbol{A}_{\mathtt{N}}\boldsymbol{H}^{(k-1)}$, where the message for node $i$, $\boldsymbol{M}_{i,:}^{(k)}$, is the average of its neighbours' embeddings.

PCAPass updates its embeddings in each iteration using a concatenation and compression approach. This update can be expressed as $\boldsymbol{H}^{(k)} = \begin{bmatrix} \boldsymbol{H}^{(k-1)} & \boldsymbol{M}^{(k)} \end{bmatrix} \boldsymbol{V}^{(k)}$, where the brackets indicate horizontal concatenation. The matrix $\boldsymbol{V}^{(k)} \in \mathbb{R}^{2p_{k-1} \times p_k}$ is derived through PCA (Murphy, 2012, Ch. 12.2) and serves to compress the embeddings. Compression is essential because, without it, the dimensionality of the embedding space would double with each message-passing iteration, resulting in a final dimension of $2^K d$. This exponential growth would impose considerable memory and computational overhead, and could negatively impact downstream tasks due to the curse of dimensionality.

Although Sadowski et al. (2022) originally formulated PCAPass for undirected graphs, it can be straightforwardly extended to directed graphs by following the approach of Rossi et al. (2023). In directed graphs, each node has two distinct sets of neighbours: those connected by incoming edges and those connected by outgoing edges. To capture the information from these distinct neighbourhoods, two separate aggregation operators are employed: $\boldsymbol{A}_{\mathtt{F}} = \boldsymbol{D}_{\mathtt{I}}^{-1}\boldsymbol{A}$ and $\boldsymbol{A}_{\mathtt{B}} = \boldsymbol{D}_{\mathtt{0}}^{-1}\boldsymbol{A}^{\mathsf{T}}$. Here, $\boldsymbol{A}_{\mathtt{F}}$ uses the adjacency matrix $\boldsymbol{A}$, while $\boldsymbol{A}_{\mathtt{B}}$ uses its transpose $\boldsymbol{A}^{\mathsf{T}}$, corresponding to a graph where all edge directions are reversed. The forward operator $\boldsymbol{A}_{\mathtt{F}}$ aggregates messages based on a node's incoming edges and normalizes by in-degree, while the backward operator $\boldsymbol{A}_{\mathtt{B}}$ aggregates based on outgoing edges and normalizes by out-degree.

With these operators, the directed aggregation step for PCAPass is defined as $\boldsymbol{M}_{\mathtt{F}}^{(k)} = \boldsymbol{A}_{\mathtt{F}}\boldsymbol{H}^{(k-1)}$ and $\boldsymbol{M}_{\mathtt{B}}^{(k)} = \boldsymbol{A}_{\mathtt{B}}\boldsymbol{H}^{(k-1)}$. In the update step, both forward and backward messages are concatenated with the previous embeddings and compressed: $\boldsymbol{H}^{(k)} = \begin{bmatrix} \boldsymbol{H}^{(k-1)} & \boldsymbol{M}_{\mathtt{F}}^{(k)} & \boldsymbol{M}_{\mathtt{B}}^{(k)} \end{bmatrix} \boldsymbol{V}^{(k)}$, with $\boldsymbol{V}^{(k)} \in \mathbb{R}^{3p_{k-1} \times p_k}$.

## 3 Embedding aggregation and concatenation results in rank deficiency

As outlined above, PCAPass (Sadowski et al., 2022) leverages embedding aggregation, concatenation, and compression to generate node embeddings. This approach aims to address the over-smoothing issue that often plagues message-passing models (Li et al., 2018; Chen et al., 2020a). However, the repeated process of embedding aggregation and concatenation introduces and retains redundant features in the embeddings. This not only leads to inefficient computation but also results in *rank deficiency*, a condition where the embedding matrix becomes ill-conditioned, adversely affecting the quality and usefulness of the embeddings.

### 3.1 Origin of rank deficiency

A matrix $\boldsymbol{Z} \in \mathbb{R}^{n \times p}$ is considered rank-deficient if it exhibits a cluster of small singular values, with a clear gap between the large and small singular values (Hansen, 1998, Ch. 1.). This situation arises when the columns of $\boldsymbol{Z}$ are not linearly independent, indicating the presence of redundant features.

To illustrate the relationship between redundant features and small singular values, consider a simple example. Let $\boldsymbol{X} \in \mathbb{R}^{n \times d}$ be a full-rank matrix with the singular value decomposition (SVD) $\boldsymbol{X} = \boldsymbol{U}\boldsymbol{\Sigma}\boldsymbol{V}^{\mathsf{T}}$. Here, $\boldsymbol{U} \in \mathbb{R}^{n \times d}$ and $\boldsymbol{V} \in \mathbb{R}^{d \times d}$ have orthogonal columns with unit norms, and $\boldsymbol{\Sigma}$ is a non-negative diagonal matrix containing the singular values in descending order (Golub & Van Loan, 2013, Ch. 2.4).

Now, define a matrix $\boldsymbol{Z} = \begin{bmatrix} \boldsymbol{X} & \boldsymbol{X} \end{bmatrix} \in \mathbb{R}^{n \times 2d}$, where each column of $\boldsymbol{X}$ is duplicated, making the last $d$ columns in $\boldsymbol{Z}$ redundant. The SVD of $\boldsymbol{Z}$ can then be expressed as $\boldsymbol{Z} = \boldsymbol{U_Z} \boldsymbol{\Sigma_Z} \boldsymbol{V_Z^\intercal}$, where

$$\boldsymbol{U_Z} = \begin{bmatrix} \boldsymbol{U} & \boldsymbol{\Upsilon_U} \end{bmatrix}, \qquad \boldsymbol{\Sigma_Z} = \begin{bmatrix} \sqrt{2}\boldsymbol{\Sigma} & \boldsymbol{0} \\ \boldsymbol{0} & \boldsymbol{0} \end{bmatrix}, \qquad \boldsymbol{V_Z^\intercal} = \frac{1}{\sqrt{2}} \begin{bmatrix} \boldsymbol{V^\intercal} & \boldsymbol{V^\intercal} \\ \boldsymbol{\Upsilon_{V_1}^\intercal} & \boldsymbol{\Upsilon_{V_2}^\intercal} \end{bmatrix}. \tag{2}$$

Here, the block matrices $\boldsymbol{\Upsilon_U} \in \mathbb{R}^{n \times d}$, $\boldsymbol{\Upsilon_{V_1}} \in \mathbb{R}^{d \times d}$, and $\boldsymbol{\Upsilon_{V_2}} \in \mathbb{R}^{d \times d}$ are orthonormal and satisfy $\boldsymbol{\Upsilon_U^\intercal} \boldsymbol{U} = \boldsymbol{0}$ and $\boldsymbol{\Upsilon_{V_1}^\intercal} \boldsymbol{V} = \boldsymbol{\Upsilon_{V_2}^\intercal} \boldsymbol{V} = \boldsymbol{0}$. We can verify this decomposition through matrix multiplication:

$$\boldsymbol{U_Z} \boldsymbol{\Sigma_Z} \boldsymbol{V_Z^\intercal} = \begin{bmatrix} \boldsymbol{U} & \boldsymbol{\Upsilon_U} \end{bmatrix} \begin{bmatrix} \sqrt{2}\boldsymbol{\Sigma} & \boldsymbol{0} \\ \boldsymbol{0} & \boldsymbol{0} \end{bmatrix} \frac{1}{\sqrt{2}} \begin{bmatrix} \boldsymbol{V^\intercal} & \boldsymbol{V^\intercal} \\ \boldsymbol{\Upsilon_{V_1}^\intercal} & \boldsymbol{\Upsilon_{V_2}^\intercal} \end{bmatrix} = \begin{bmatrix} \boldsymbol{U\Sigma} & \boldsymbol{0} \end{bmatrix} \begin{bmatrix} \boldsymbol{V^\intercal} & \boldsymbol{V^\intercal} \\ \boldsymbol{\Upsilon_{V_1}^\intercal} & \boldsymbol{\Upsilon_{V_2}^\intercal} \end{bmatrix} = \begin{bmatrix} \boldsymbol{U\Sigma V^\intercal} & \boldsymbol{U\Sigma V^\intercal} \end{bmatrix}. \tag{3}$$

(See Appendix A for verification of the orthonormality of $\boldsymbol{U_Z}$ and $\boldsymbol{V_Z}$.)

The expression for $\boldsymbol{\Sigma_Z}$ in Equation 2 reveals two clusters of singular values: the $d$ values contained in $\boldsymbol{\Sigma}$ and $d$ singular values equal to zero. Each of the zero singular values corresponds to a column in $\boldsymbol{\Upsilon_U}$, which we refer to as singular dimensions. These singular dimensions span the left null space of $\boldsymbol{Z}$ and do not contain any information about $\boldsymbol{X}$. All information about $\boldsymbol{X}$ is already encapsulated in $\boldsymbol{U}$, $\boldsymbol{\Sigma}$, and $\boldsymbol{V}$, as demonstrated by Equation 3. The columns of $\boldsymbol{\Upsilon_U}$ are only constrained by their orthogonality to the columns in $\boldsymbol{U}$, and their elements can otherwise be chosen arbitrarily. Similarly, $\boldsymbol{\Upsilon_{V_1}}$ and $\boldsymbol{\Upsilon_{V_2}}$ are only constrained by their orthogonality to $\boldsymbol{V}$.

Although the values of $\boldsymbol{\Upsilon_U}$, $\boldsymbol{\Upsilon_{V_1}}$, and $\boldsymbol{\Upsilon_{V_2}}$ are not uniquely defined, in practical applications, they are determined by a combination of numerical inaccuracies and the arbitrary implementation choices of the specific algorithm used to compute the SVD. Since they do not carry any meaningful information about $\boldsymbol{X}$, they can effectively be considered noise. Furthermore, numerical implementations of SVD will not produce singular values that are exactly zero due to rounding errors, as reflected in the definition of rank deficiency.

### 3.2 Rank deficiency in PCAPass

Having established the connection between redundant features and rank deficiency, we now analyse the message-passing mechanism in PCAPass, as described in Section 2, to show how redundant features contribute to rank deficiency in its embeddings. For simplicity, we focus on undirected message-passing; however, the results readily extend to directed graphs.

Let $p_k$ denote the dimension of the embeddings after compression in the $k$th iteration, with $p_0 = d$, and let $\boldsymbol{W}^{(k)} \in \mathbb{R}^{2p_{k-1} \times p_k}$ denote the compression matrix for iteration $k$. Then, the PCAPass message-passing update step can be expressed as $\boldsymbol{H}^{(k)} = \begin{bmatrix} \boldsymbol{H}^{(k-1)} & \boldsymbol{A_N} \boldsymbol{H}^{(k-1)} \end{bmatrix} \boldsymbol{W}^{(k)}$, with the first two iterations given by

$$\begin{aligned} \boldsymbol{H}^{(1)} &= \begin{bmatrix} \boldsymbol{X} & \boldsymbol{A_N} \boldsymbol{X} \end{bmatrix} \boldsymbol{W}^{(1)}, \\ \boldsymbol{H}^{(2)} &= \begin{bmatrix} \boldsymbol{H}^{(1)} & \boldsymbol{A_N} \boldsymbol{H}^{(1)} \end{bmatrix} \boldsymbol{W}^{(2)}. \end{aligned} \tag{4}$$

By inserting $\boldsymbol{H}^{(1)}$ into the expression for $\boldsymbol{H}^{(2)}$, and factoring out $\boldsymbol{W}^{(1)}$, we obtain the following result:

$$\begin{aligned} \boldsymbol{H}^{(2)} &= \begin{bmatrix} \boldsymbol{H}^{(1)} & \boldsymbol{A_N} \boldsymbol{H}^{(1)} \end{bmatrix} \boldsymbol{W}^{(2)} \\ &= \begin{bmatrix} \begin{bmatrix} \boldsymbol{X} & \boldsymbol{A_N} \boldsymbol{X} \end{bmatrix} \boldsymbol{W}^{(1)} & \boldsymbol{A_N} \begin{bmatrix} \boldsymbol{X} & \boldsymbol{A_N} \boldsymbol{X} \end{bmatrix} \boldsymbol{W}^{(1)} \end{bmatrix} \boldsymbol{W}^{(2)} \\ &= \begin{bmatrix} \boldsymbol{X} & \boldsymbol{A_N} \boldsymbol{X} & \boldsymbol{A_N} \boldsymbol{X} & \boldsymbol{A_N^2} \boldsymbol{X} \end{bmatrix} \begin{bmatrix} \boldsymbol{W}^{(1)} & \boldsymbol{0} \\ \boldsymbol{0} & \boldsymbol{W}^{(1)} \end{bmatrix} \boldsymbol{W}^{(2)} \\ &= \boldsymbol{A_X^{(2)}} \hat{\boldsymbol{W}}^{(2)}, \end{aligned} \tag{5}$$

$$\boldsymbol{A_X^{(2)}} = \begin{bmatrix} \boldsymbol{X} & \boldsymbol{A_N} \boldsymbol{X} & \boldsymbol{A_N} \boldsymbol{X} & \boldsymbol{A_N^2} \boldsymbol{X} \end{bmatrix} \in \mathbb{R}^{n \times 4d}, \tag{6}$$

$$\hat{\boldsymbol{W}}^{(2)} = \begin{bmatrix} \boldsymbol{W}^{(1)} & \boldsymbol{0} \\ \boldsymbol{0} & \boldsymbol{W}^{(1)} \end{bmatrix} \boldsymbol{W}^{(2)} \in \mathbb{R}^{4d \times p_2}. \tag{7}$$

We see that $\boldsymbol{H}^{(2)} = \boldsymbol{A}_{\boldsymbol{X}}^{(2)} \hat{\boldsymbol{W}}^{(2)}$, where $\boldsymbol{A}_{\boldsymbol{X}}^{(2)}$ consists of four submatrices of aggregated node features. We can express $\boldsymbol{H}^{(3)}$ on the same block matrix form:

$$
\begin{aligned}
\boldsymbol{H}^{(3)} &= \begin{bmatrix} \boldsymbol{H}^{(2)} & \boldsymbol{A}_{\mathbb{N}} \boldsymbol{H}^{(2)} \end{bmatrix} \boldsymbol{W}^{(3)} \\
&= \begin{bmatrix} \boldsymbol{X} & \boldsymbol{A}_{\mathbb{N}} \boldsymbol{X} & \boldsymbol{A}_{\mathbb{N}} \boldsymbol{X} & \boldsymbol{A}_{\mathbb{N}}^2 \boldsymbol{X} & \boldsymbol{A}_{\mathbb{N}} \boldsymbol{X} & \boldsymbol{A}_{\mathbb{N}}^2 \boldsymbol{X} & \boldsymbol{A}_{\mathbb{N}}^2 \boldsymbol{X} & \boldsymbol{A}_{\mathbb{N}}^3 \boldsymbol{X} \end{bmatrix} \begin{bmatrix} \hat{\boldsymbol{W}}^{(2)} & \boldsymbol{0} \\ \boldsymbol{0} & \hat{\boldsymbol{W}}^{(2)} \end{bmatrix} \boldsymbol{W}^{(3)} \qquad (8) \\
&= \boldsymbol{A}_{\boldsymbol{X}}^{(3)} \hat{\boldsymbol{W}}^{(3)},
\end{aligned}
$$

$$
\boldsymbol{A}_{\boldsymbol{X}}^{(3)} = \begin{bmatrix} \boldsymbol{X} & \boldsymbol{A}_{\mathbb{N}} \boldsymbol{X} & \boldsymbol{A}_{\mathbb{N}} \boldsymbol{X} & \boldsymbol{A}_{\mathbb{N}}^2 \boldsymbol{X} & \boldsymbol{A}_{\mathbb{N}} \boldsymbol{X} & \boldsymbol{A}_{\mathbb{N}}^2 \boldsymbol{X} & \boldsymbol{A}_{\mathbb{N}}^2 \boldsymbol{X} & \boldsymbol{A}_{\mathbb{N}}^3 \boldsymbol{X} \end{bmatrix} \in \mathbb{R}^{n \times 8d}, \qquad (9)
$$

$$
\hat{\boldsymbol{W}}^{(3)} = \begin{bmatrix} \hat{\boldsymbol{W}}^{(2)} & \boldsymbol{0} \\ \boldsymbol{0} & \hat{\boldsymbol{W}}^{(2)} \end{bmatrix} \boldsymbol{W}^{(3)} \in \mathbb{R}^{8d \times p_3}. \qquad (10)
$$

Continuing this pattern, we see that $\boldsymbol{H}^{(k)} = \boldsymbol{A}_{\boldsymbol{X}}^{(k)} \hat{\boldsymbol{W}}^{(k)}$, where $\boldsymbol{A}_{\boldsymbol{X}}^{(k)} \in \mathbb{R}^{n \times 2^k d}$ and $\hat{\boldsymbol{W}}^{(k)} \in \mathbb{R}^{2^k d \times p_k}$.

Notice that the submatrix $\boldsymbol{A}_{\mathbb{N}} \boldsymbol{X}$ is repeated twice in $\boldsymbol{A}_{\boldsymbol{X}}^{(2)}$ and three times in $\boldsymbol{A}_{\boldsymbol{X}}^{(3)}$, where $\boldsymbol{A}_{\mathbb{N}}^2 \boldsymbol{X}$ also appears three times. These redundant submatrices contribute to the rank deficiency of the PCAPass embeddings. In fact, only one linearly independent submatrix is introduced in each message-passing iteration, specifically $\boldsymbol{A}_{\mathbb{N}}^k \boldsymbol{X}$ in iteration $k$, implying that $\operatorname{rank}(\boldsymbol{A}_{\boldsymbol{X}}^{(k)}) \leq (k+1)d$. Consequently, we can obtain an upper bound on the rank of $\boldsymbol{H}^{(k)}$ by applying Sylvester's inequality (Horn & Johnson, 2012, Ch. 0.4.5):

$$
\operatorname{rank}(\boldsymbol{H}^{(k)}) \leq \min(\operatorname{rank}(\boldsymbol{A}_{\boldsymbol{X}}^{(k)}), \operatorname{rank}(\hat{\boldsymbol{W}}^{(k)})) \leq \operatorname{rank}(\boldsymbol{A}_{\boldsymbol{X}}^{(k)}) \leq (k+1)d. \qquad (11)
$$

Thus, we have $\rho_k \geq p_k - (k+1)d$, with $\rho_k$ being the number of singular values close to zero for $\boldsymbol{H}^{(k)} \in \mathbb{R}^{n \times p_k}$. This bound indicates that the PCAPass embeddings will be rank-deficient unless $p_k \leq (k+1)d$. However, note that setting $p_k = (k+1)d$ does not guarantee full-rank embeddings, as additional sources of rank deficiency may be present, such as the potential rank deficiency of $\boldsymbol{X}$ itself.

Having established the rank-deficiency of the PCAPass embeddings regardless of the specific compression method used to compute each $\boldsymbol{W}^{(k)}$, we now examine the PCA compression utilized by Sadowski et al. (2022) in greater detail. Recall that PCA is typically and efficiently implemented through centring and SVD (Murphy, 2012, Ch. 12.2.3). Therefore, if all singular dimensions are retained, the final PCAPass embeddings can be expressed as $\boldsymbol{Z} = \begin{bmatrix} \boldsymbol{U}\boldsymbol{\Sigma} & \boldsymbol{\Upsilon}\boldsymbol{\Sigma}_{\approx 0} \end{bmatrix}$ Here, $\boldsymbol{U}\boldsymbol{\Sigma} \in \mathbb{R}^{n \times (k+1)d}$ represents the informative part of the embeddings, while $\boldsymbol{\Upsilon}\boldsymbol{\Sigma}_{\approx 0} \in \mathbb{R}^{n \times \rho_k}$ captures the singular dimensions, with elements in $\boldsymbol{\Sigma}_{\approx 0}$ being close to zero.

Applying compression in each iteration has the potential to address the rank deficiency by removing the singular dimensions $\boldsymbol{\Upsilon}$. However, the effectiveness of this approach depends on the specifics of the PCA compression. Sadowski et al. (2022) implement a maximum embedding dimension, $p_{\max}$, such that $p_k = \min(p_{\max}, 2^k d)$, where $2^k d$ is the embedding dimension without compression for undirected message-passing. This implies that the PCAPass embedding will remain rank-deficient in each iteration until $(k+1)d \geq p_{\max}$. Thus, using a fixed value for $p_{\max}$ does not ensure the removal of singular dimensions. Instead, the number of retained dimensions would need to vary with each iteration $k$ to guarantee the elimination of all singular dimensions. This introduces additional algorithmic complexity, especially since the bound on $p_k$ must be adapted for directed message-passing and for message-passing with multiple aggregation operations (Jin et al., 2019; Corso et al., 2020).

An alternative approach is to apply a threshold to the singular values during compression, removing any dimensions where $\Sigma_{i,i} \leq \theta \Sigma_{1,1}$, where $\Sigma_{1,1}$ is the largest singular value and $\theta \in [0,1]$ a relative tolerance. However, setting $\theta$ either too low or too high can lead to a loss of accuracy in downstream tasks, see Section 5.1.

Moreover, even if the singular dimensions are successfully eliminated, computational resources have still been unnecessarily spent on computing the redundant features in the first place. Given that the number of singular dimensions can grow rapidly, as suggested by the bound $\rho_k \geq (2^k - k - 1)d$, and that PCA compression has quadratic time complexity with respect to the number of features (Golub & Van Loan, 2013, Ch. 2.4), the amount of unnecessary computation is substantial.

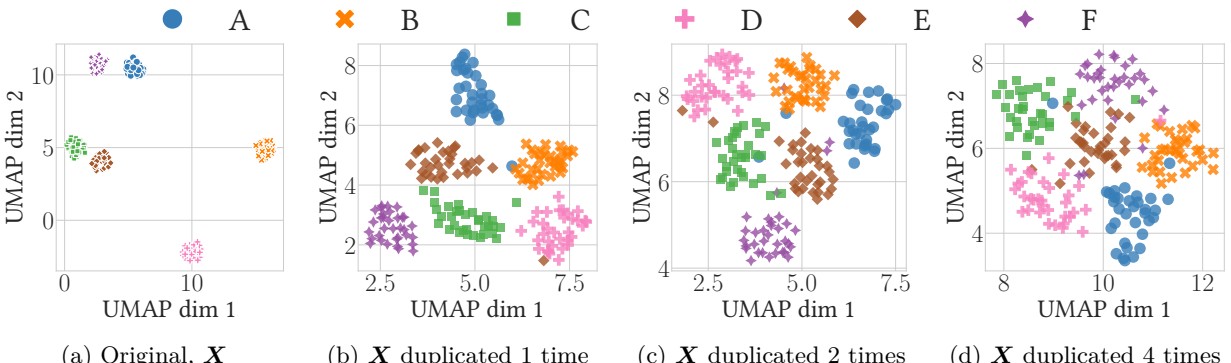

(a) Original, $\boldsymbol{X}$     (b) $\boldsymbol{X}$ duplicated 1 time     (c) $\boldsymbol{X}$ duplicated 2 times     (d) $\boldsymbol{X}$ duplicated 4 times

Figure 2: The effect of increasing rank deficiency on cluster structures. Figure 2a shows UMAP projection of $\boldsymbol{X}$, comprising 200 data points from 6 classes originally sampled on the surface of a 6D sphere. Figures 2a to 2d illustrate the cluster structure deterioration as $\boldsymbol{X}$ is horizontally concatenated with itself and whitened.

### 3.3 The negative effects of rank-deficient embeddings on clustering

Rank-deficient or otherwise ill-conditioned embedding matrices can violate fundamental assumptions made by downstream machine learning models, like the linear independence of features, and contribute to the instability of numerical computations (Trefethen & Bau, 1997, Pt. 3). In supervised learning, the adverse effects of rank deficiency are often mitigated through regularization techniques. For example, ridge regression (Hoerl & Kennard, 1970) effectively addresses this issue. However, regularization and model selection become much more challenging in unsupervised settings, where the absence of a guiding supervision signal complicates the process (Ma et al., 2023). As a result, unsupervised tasks that rely on embedding distances, such as clustering and graph alignment, are particularly vulnerable to the negative impacts of rank deficiency.

The rank-deficient embedding matrices produced by PCAPass are especially problematic when subjected to common preprocessing operations like standardization (Aggarwal, 2015, Ch. 2.3.3) and whitening (Hyvärinen et al., 2009). We illustrate this with a simple clustering example. We generate a matrix $\boldsymbol{X} \in \mathbb{R}^{200 \times 6}$, consisting of 200 six-dimensional samples. The data is divided into six classes, each corresponding to a normal distribution centred at one of six equidistant points. This setup ensures that samples from each Gaussian blob are well-separated. We visualize this in Figure 2a using 2D UMAP projections (McInnes et al., 2018).

To simulate the introduction of redundant features, we concatenate $\boldsymbol{X}$ with itself, forming $\tilde{\boldsymbol{X}} = \begin{bmatrix} \boldsymbol{X} & \boldsymbol{X} \end{bmatrix}$. Applying PCA-based whitening to $\tilde{\boldsymbol{X}}$ yields the simulated embedding matrix $\boldsymbol{Z} = \sqrt{n-1} \begin{bmatrix} \boldsymbol{U} & \boldsymbol{\Upsilon} \end{bmatrix}$. This expression highlights the connection to PCAPass embeddings that have been standardized to unit variance.

Figure 2 visually demonstrates how cluster separability diminishes as the number of duplicate features increases. This degradation in separability can also be quantified by running k-means clustering (Arthur & Vassilvitskii, 2007). The average normalized mutual information (NMI) (Danon et al., 2005) over 10 seeds is 1.0 for the original data, indicating perfect separation of the classes. However, as $\boldsymbol{X}$ is duplicated, the NMI decreases to 0.78, and then to 0.71 with two duplicates, and down to 0.47 with four duplicates.

Note that the information required to distinguish the clusters remains present in $\boldsymbol{Z}$, and a regularized supervised classifier could still achieve high classification accuracy. However, the relevant information has been diluted by the singular dimensions, $\boldsymbol{\Upsilon}$, which significantly impairs the class's clustering separability.

In this toy example, the negative effects of the singular dimensions can be mitigated by using $\boldsymbol{Z} = \begin{bmatrix} \boldsymbol{U\Sigma} & \boldsymbol{\Upsilon\Sigma_{\approx 0}} \end{bmatrix}$ as input for clustering, which simulates not preprocessing the PCAPass embeddings. However, in practice, preprocessing is often necessary for achieving high task accuracy. Embedding-based graph alignment using structural input features is one such example.

### 3.4 The negative effects of rank-deficient embeddings on graph alignment

Graph alignment is the task of finding node correspondences between two graphs, $\mathcal{G}_1 = (\mathbb{N}, \mathbb{M}_1)$ and $\mathcal{G}_2 = (\mathbb{N}, \mathbb{M}_2)$. We assume, for simplicity, that while the node set $\mathbb{N}$ is shared, while the edge sets differ. When $\mathbb{M}_2$ is a subset or superset of $\mathbb{M}_1$, we refer to the missing or added edges as noise edges.

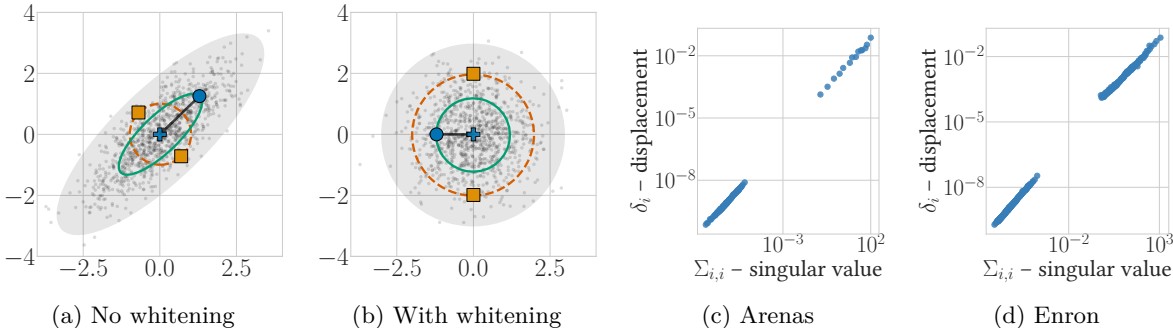

Figure 3: Visualization explaining the need for whitening in embedding-based graph alignment. In Figure 3a, the goal is to match the blue plus with the blue circle. However, the Euclidean distance fails to account for the data's shape, causing the erroneous orange squares to appear closer. Whitening corrects this issue, as shown in Figure 3b. The assumption in Figure 3a is that the direction of displacement aligns with the data's variation. Figures 3c and 3d provide empirical evidence of a strong correlation between the displacement along axis $i$ and the corresponding singular value, $\Sigma_{i,i}$, based on PCAPass embeddings.

Embedding-based graph alignment (Heimann et al., 2018) is a greedy approach where node embeddings are used to match the nodes. Specifically, the nodes in $\mathcal{G}_2$ are matched to the nodes in $\mathcal{G}_1$ with the most similar embeddings in terms of Euclidean distance, which can be computed efficiently using KD-trees (Bentley, 1975). When the graphs lack node attributes, *structural node embeddings* are used (Jin et al., 2021). These embeddings can be generated by message-passing models where $\boldsymbol{X}$ consists of structural features such as node degrees and local clustering coefficients (Fagiolo, 2007).

For this approach to be effective, the embeddings need to be whitened before matching. This is detrimental to the alignment accuracy of PCAPass embeddings, since applying whitening removes the scaling of $\boldsymbol{\Sigma}_{\approx 0}$ from the singular dimensions, allowing $\boldsymbol{\Upsilon}$ to introduce noise into the distance computation. Consequently, the PCAPass embeddings struggle to produce high graph alignment accuracies, as we demonstrate in Section 5.

Figure 3 provides a visual explanation for why whitening is necessary. The gray scatter points in Figure 3a represent the embeddings for each node in $\mathcal{G}_1$, and the ellipse highlights variations along the principal axes of the embedding matrix. The coloured points show a magnified version of the alignment process. The blue circle represents an embedding vector in $\mathcal{G}_1$, and the blue plus represents the corresponding node in $\mathcal{G}_2$. The orange squares represent embeddings for other nodes in $\mathcal{G}_1$.

As shown, both orange squares are closer to the blue plus than the blue circle in terms of Euclidean distance. Thus, this node would be incorrectly matched. However, if the covariance of the data is considered in the

---

**Algorithm 1:** The ACC algorithm for a directed graph $\mathcal{G}$ with node features $\boldsymbol{X}$ using $K$ message-passing iterations, desired dimensionality $p_{\max}$, minimal message size $c_{\min}$ and relative tolerance $\theta \in [0,1]$.

```
1  def ACC(𝒢, X, K, p_max, c_min = 2, θ = 10⁻⁸):
2      c = max(⌊p_max/(K + 1)⌋, c_min)
3      M⁽⁰⁾ = PCA(X, c, θ)
4      H⁽⁰⁾ = M⁽⁰⁾
5      for k in range(1, K + 1):
          # Compute aggregations in both direction.
6          M⁽ᵏ⁾ = [A_F M⁽ᵏ⁻¹⁾   A_B M⁽ᵏ⁻¹⁾]
7          M⁽ᵏ⁾ ← PCA(M⁽ᵏ⁾, c, θ)
          # Concatenate with existing embeddings.
8          H⁽ᵏ⁾ = [H⁽ᵏ⁻¹⁾   M⁽ᵏ⁾]
9      return H⁽ᴷ⁾
```

**Algorithm 2:** PCA compression computed via SVD as used for ACC. Here $\boldsymbol{X} \in \mathbb{R}^{n \times p}$ is a real-valued matrix, $c$ an integer s.t. $c \leq p$, and $\theta \in [0,1]$ a relative tolerance.

```
1  def PCA(X, c, θ):
2      X ← centre_columns(X)
3      U, Σ, Vᵀ = SVD(X)
       # Find the index of the smallest
         singular value which exceeds the
         given tolerance relative to the
         largest singular value Σ₁,₁.
4      k_θ = max{j | Σ_{j,j} ≥ θΣ₁,₁}
       # Keep at most k_θ dimensions.
5      k = min(c, k_θ)
6      X_c = XV_{:,:k}
7      return X_c
```

distance calculation, the blue plus is closer to the blue circle, as indicated by the green ellipse. Whitening spheres the data, equalizing the variance along each principal axis. Therefore, Euclidean distance provides the correct matching in the whitened space, as shown in Figure 3b.

This explanation assumes that the direction of the embedding displacement, indicated by the black arrow in Figure 3a, correlates with the data variation. Figures 3c and 3d provide empirical support for this assumption. These figures show the singular values of the PCAPass embeddings for $\mathcal{G}_1$, denoted $\boldsymbol{Z}^{(1)}$, on the x-axes. The y-axes show the average embedding displacement along the corresponding principal axis, defined as $\delta_i = \frac{1}{n}\|(\boldsymbol{Z}^{(1)} - \boldsymbol{Z}^{(2)})\boldsymbol{V}_{:,i}\|_2$, where $\boldsymbol{V}$ is a basis for the principal axes of $\boldsymbol{Z}^{(1)}$. As seen, the correlation between the directions of variation and displacement is strong, supporting our assumption.

In summary, the rank-deficient embeddings produced by PCAPass contain arbitrary column vectors that act as noise for unsupervised downstream tasks such as clustering and graph alignment. This issue is particularly problematic when normalizing preprocessing steps are required for optimal performance, as is the case with whitening for graph alignment. Therefore, it is crucial to avoid rank-deficient node embeddings.

## 4 The ACC model and message aggregation

To address the issues of rank-deficient embeddings, we introduce the ACC, which stands for **A**ggregate, **C**ompress, and **C**oncatenate. Performed in this order, these operations result in a message-passing approach where the message matrices, rather than the embeddings, are passed between message-passing iterations. We refer to this as *message aggregation*, and as we will demonstrate, it avoids computing the redundant features that cause the rank deficiency observed in PCAPass. Furthermore, ACC is designed to work with directed graphs, expanding its applicability and versatility.

The complete ACC algorithm is detailed in pseudocode in Algorithm 1. Initially, the feature matrix $\boldsymbol{X}$ is compressed into a $c$-dimensional message matrix, $\boldsymbol{M}^{(0)} \in \mathbb{R}^{n \times c}$, using Principal Component Analysis (PCA) (Murphy, 2012, Ch. 12.2). Once this initial compression is complete, the message-passing procedure begins. The main distinguishing feature of ACC, as shown on Line 6 of Algorithm 1, is *message aggregation*. This means that ACC applies aggregation operators directly to the message matrices from the previous iteration, $\boldsymbol{M}^{(k)} = \begin{bmatrix} \boldsymbol{A}_{\mathrm{F}}\boldsymbol{M}^{(k-1)} & \boldsymbol{A}_{\mathrm{B}}\boldsymbol{M}^{(k-1)} \end{bmatrix}$. This approach is crucial for avoiding the creation of redundant features, as seen in PCAPass when aggregating and concatenating embedding matrices in Equation 9.

The new message matrix $\boldsymbol{M}^{(k)}$ is then compressed to $c$ dimensions using PCA and concatenated with the embedding matrix, $\boldsymbol{H}^{(k)} = \begin{bmatrix} \boldsymbol{H}^{(k-1)} & \boldsymbol{M}^{(k)} \end{bmatrix}$. Consequently, the final embedding matrix is a concatenation of all message matrices, $\boldsymbol{H}^{(K)} = \begin{bmatrix} \boldsymbol{M}^{(0)} & \boldsymbol{M}^{(1)} \dots & \boldsymbol{M}^{(K)} \end{bmatrix}$, with a total embedding dimension of $p = (K+1)c$.

Typically, the value of $c$ is chosen as the largest integer such that the total embedding dimension $p$ does not exceed a desired limit, $(K + 1)c \le p_{\max}$. However, this condition would result in $c = 0$ if $(K + 1) > p_{\max}$. To prevent this, ACC enforces a minimal value for $c$, denoted as $c_{\min}$, as specified on Line 2 of Algorithm 1. By default, we set $c_{\min} = 2$ to account for the two message-passing directions in directed graphs.

For undirected graphs, ACC is slightly modified on Line 6, as message-passing is conducted using only the undirected matrix $\boldsymbol{A}_{\mathbb{N}}$, $\boldsymbol{M}^{(k)} = \boldsymbol{A}_{\mathbb{N}}\boldsymbol{M}^{(k-1)}$. In this scenario, the compression steps on Line 7 is technically unnecessary since $\boldsymbol{M}^{(k)}$ is already $c$-dimensional. Nonetheless, for consistency across our experiments, we always perform PCA.

To demonstrate that ACC avoids computing the redundant features seen for PCAPass in Equation 9, we write out the expression for the undirected ACC embeddings after $K$ iterations: $\boldsymbol{H}^{(K)} = \begin{bmatrix} \boldsymbol{X}\boldsymbol{W}^{(0)}, \boldsymbol{A}_{\mathbb{N}}\boldsymbol{X}\boldsymbol{W}^{(1)}, \dots, \boldsymbol{A}_{\mathbb{N}}^K \boldsymbol{X}\boldsymbol{W}^{(K)} \end{bmatrix}$. In this formulation, each $\boldsymbol{W}^{(k)} \in \mathbb{R}^{d \times c}$ is a compression matrix derived from successive applications of PCA. Unlike PCAPass, the ACC embeddings exclude the repeated submatrices found in Equation 9. Specifically, $\boldsymbol{H}^{(K)}$ in ACC contains exactly one instance of $\boldsymbol{A}_{\mathbb{N}}^k \boldsymbol{X}$ for each $k \in \{0, \dots, K\}$, representing the non-redundant information produced at each iteration.

We can further verify that the ACC embeddings are not rank-deficient by examining the number of singular values close to zero, denoted as $\rho_K$, where $K$ represents the number of message-passing iterations. To compute $\rho_K$, we count the number of singular values for which $\Sigma_{ii} \le 10^{-6}\Sigma_{11}$. Figure 4 illustrates $\rho_K$ for

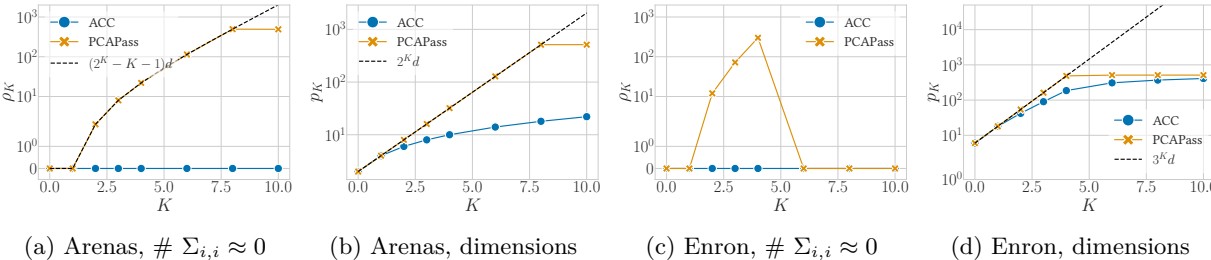

(a) Arenas, $\# \Sigma_{i,i} \approx 0$      (b) Arenas, dimensions      (c) Enron, $\# \Sigma_{i,i} \approx 0$      (d) Enron, dimensions

Figure 4: Figures 4a and 4c show the number of small singular values, $\rho_K$, as a function of the number of message-passing iterations, $K$, for two graphs: Arenas and Enron. The dashed line in 4a represents the theoretical prediction for $\rho_K$ discussed in Section 3.2. Figures 4b and 4d display the embedding dimensionality $p_K$. Both ACC and PCAPass use the maximum dimensionality $p_{\max} = 512$, resulting in the curve cut-off.

both the ACC and PCAPass embeddings, along with the corresponding embedding dimensions $p_K$, for two graphs: Arenas, an undirected graph, and Enron, a directed graph. The initial number of features is $d = 2$ for Arenas and $d = 6$ for Enron. In both cases, the maximum embedding dimensionality $p_{\max} = 512$ is applied.

We observe that the number of small singular values remains at zero for all values of $K$ for ACC, indicating that its embeddings are full rank. In contrast, for PCAPass, $\rho_K$ grows rapidly for both graphs until the embedding dimensionality $p_K$ reaches the maximum value $p_{\max}$. For Arenas, the growth of $\rho_K$ closely follows our predicted lower bound $(2^k - k - 1)d$, as depicted by the dashed black curve in Figure 4a. For Enron, $\rho_K$ initially increases more rapidly than for Arenas, consistent with the fact that the dimensionality $p_K$ expands more quickly for directed graphs due to the concatenation of forward and backward aggregations. This also results in $p_{\max}$ being reached sooner, and as a consequence, the singular dimensions are replaced by non-redundant features after $K = 4$, causing $\rho_K$ to drop to zero. This analysis highlights the interaction between the PCAPass rank deficiency, the number of message-passing iterations, and the maximum dimensionality used for PCA compression.

While message aggregation effectively avoids computing the redundant features highlighted for PCAPass in Equation 8, other sources of singular dimensions can still arise. For instance, singular dimensions can be introduced if $\boldsymbol{X}$ itself is not full rank, or if the adjacency matrix contains eigenvalues close to zero. To mitigate these issues, ACC employs a small threshold $\theta$ during the PCA compression to discard embedding dimensions with small singular values, as outlined in Lines 4 to 6 of Algorithm 2.

However, singular dimensions can also emerge due to correlations across message-passing iterations. For example, if $\boldsymbol{X}$ contains an eigenvector of $\boldsymbol{A}_\mathbb{N}$, this would introduce a singular dimension in $\begin{bmatrix} \boldsymbol{X} & \boldsymbol{A}_\mathbb{N}\boldsymbol{X} \end{bmatrix}$. ACC cannot remove these correlations during message-passing because the messages from each iteration are compressed independently. This further implies that the dimensions of the final ACC embeddings, $\boldsymbol{Z}$, may exhibit correlations, which could impact certain downstream tasks.

To address these concerns, decorrelation and removal of singular dimensions from $\boldsymbol{H}^{(K)}$ after message-passing could be beneficial. However, in our experiments, we have not incorporated these steps by default, treating them instead as potential preprocessing steps for specific downstream tasks.

## 4.1 Time complexity, scalability, and the inductive learning setting

The time complexity of ACC is primarily determined by the message aggregation and PCA compression steps (lines 3, 6, and 7 in Algorithm 1). For an input matrix of size $n \times p$, PCA has complexity $\mathcal{O}(np^2)$, arising from both the complexity of SVD (Golub & Van Loan, 2013, Ch. 5.5.6) and the matrix multiplication in Algorithm 2, line 6. Thus, line 3 has complexity $\mathcal{O}(nd^2)$. The message aggregation step (line 6) involves sparse-dense matrix multiplications with complexity $\mathcal{O}(mc_k)$, where $c_k$ denotes the dimension of the message matrix $\boldsymbol{M}^{(k-1)}$ from the previous iteration. The PCA compression step (line 7) has complexity $\mathcal{O}(nc_k^2)$, which also varies with iteration $k$. Using the bound $c_k \leq \frac{p_{\max}}{K+1}$ from line 2, we derive the overall worst-case complexity for $K$ iterations as $\mathcal{O}\left(nd^2 + \frac{K}{K+1}mp_{\max} + \frac{K}{(K+1)^2}np_{\max}^2\right)$.

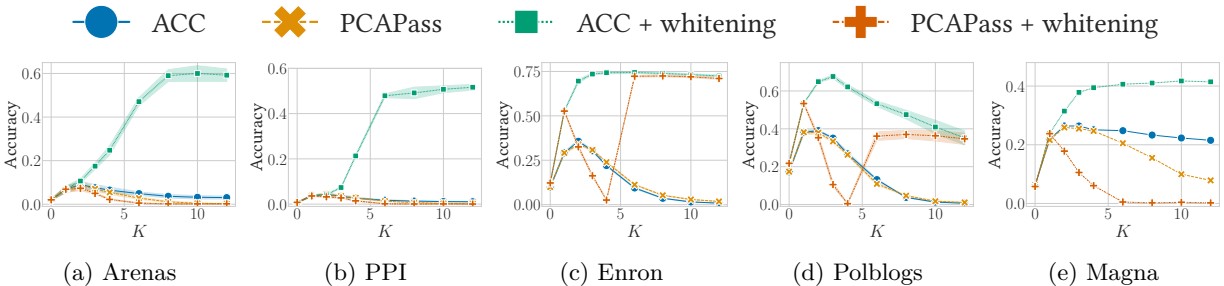

(a) Arenas  (b) PPI  (c) Enron  (d) Polblogs  (e) Magna

Figure 5: ACC and PCAPass for graph alignment with 15% noise edges. The x-axis shows the number of message-passing iterations $K$. Markers and shaded areas represent the average and standard deviation over 5 seeds. Without whitening, both algorithms perform poorly. Whitening benefits ACC across all datasets, with its accuracy reaching 60% on Arenas compared to 10% for PCAPass. Whitening only benefits PCAPass when embeddings are not rank-deficient, as shown in Figure 4.

The linear dependence on $n$ and $m$ highlights ACC's scalability. Furthermore, both message aggregation and PCA compression benefit from parallelization and GPU acceleration, as supported by prevalent graph learning libraries (Paszke et al., 2019; Fey & Lenssen, 2019). Importantly, ACC learns embeddings in a single forward pass without gradient-based optimization. This is in contrast to self-supervised graph neural networks, which typically require hundreds or thousands of training epochs (Hou et al., 2022), making ACC a computationally efficient alternative.

PCAPass similarly features linear complexity in $n$ and $m$. However, while ACC restricts message matrix dimensionality with $c_k \leq \frac{p_{\max}}{K+1}$, the corresponding bound for PCAPass is $c_k \leq p_{\max}$. Consequently, the worst-case complexity for PCAPass over $K$ iterations is $\mathcal{O}(Kmp_{\max} + Knp_{\max}^2)$ which represents an increase by a factor of $K+1$ in the term involving $m$, and by a factor of $(K+1)^2$ in the term involving $n$, compared to ACC.

ACC is primarily designed for transductive learning, where the entire graph is available during training. Given its efficiency, recomputation of embeddings upon graph updates remains straightforward. However, ACC can also be adapted for inductive learning using a GraphSAGE-inspired approach (Hamilton et al., 2017). Given a new node with known features and connections to existing nodes, its embedding can be computed by aggregating messages from neighbors within a sampled subgraph.

The primary challenge for inductive learning with ACC is the PCA-based compression step, since the projection matrices $V^{(k)}$ are derived from the original training graph. One practical solution is to project new message matrices $M^{(k)}$ onto the previously computed PCA bases, ensuring consistency with existing embeddings. Additionally, incremental PCA methods could be employed to dynamically update the bases without full recomputation (Balsubramani et al., 2013; Yu et al., 2017).

Both subgraph sampling and incremental PCA introduce potential sources of noise, necessitating careful empirical evaluation. Given ACC's computational efficiency, inductive extensions are primarily relevant for very large graphs, and we thus leave a detailed empirical investigation of such extensions to future work.

## 5 Experiments

In this section, we empirically demonstrate the superior graph alignment accuracy of the ACC model compared to PCAPass, highlighting the negative impact of rank deficiency. Additionally, we compare ACC to state-of-the-art self-supervised graph neural networks (SSGNNs) across five standard node classification benchmarks for directed graphs (Rossi et al., 2023). Our results show that ACC is over 70 times faster on the largest datasets while achieving better accuracy with default hyperparameters compared to the SSGNNs.

### 5.1 Graph alignment: ACC vs. PCAPass

To evaluate alignment accuracy, we adopt the established experimental protocol from Heimann et al. (2018). Specifically, we generate a second graph $\mathcal{G}_2$ from a reference graph $\mathcal{G}_1$ by permuting node indices and removing 15% of the edges. We conduct experiments using four datasets: Arenas, PPI, Enron, and Polblogs. Arenas

and PPI are widely used undirected benchmark graphs (Heimann et al., 2018; Jin et al., 2021; Skitsas et al., 2023), while Enron and Polblogs provide examples of directed graphs.

Additionally, we evaluate performance on the real-world Magna dataset (Saraph & Milenković, 2014), where the perturbed graph $\mathcal{G}_2$ contains 15% *more* edges than $\mathcal{G}_1$. These noisy edges were selected by Saraph & Milenković (2014) based on observed node interaction probabilities. Real-world datasets like Magna are rare due to the complexity inherent to the graph alignment problem. Detailed dataset statistics are provided in Appendix C.1.

For both ACC and PCAPass, we set the maximum embedding dimension to $p_{\max} = 512$ and use node structural features as input, $\boldsymbol{X} \in \mathbb{R}^{n \times d}$. For undirected graphs, the input features are the node degree and local clustering coefficient ($d = 2$). For directed graphs, we expand the feature set to include the in-degree, out-degree, and the four local clustering coefficients—*out*, *in*, *cycle*, and *middleman*—as defined by Fagiolo (2007), resulting in $d = 6$.

We first compare ACC and PCAPass across varying numbers of message-passing iterations $K$, with and without whitening, as shown in Figure 5. Without whitening, both ACC and PCAPass yield low accuracy, as expected given the necessity of whitening for embedding-based graph alignment (see discussion in Section 3.4). With whitening, ACC accuracy markedly improves on all datasets, reaching 60% on Arenas and 75% on Enron. In contrast, PCAPass accuracy remains low for undirected graphs, peaking at only 10% on Arenas. For directed graphs, accuracy initially increases, then deteriorates, and eventually recovers. This fluctuation is closely linked to the presence of singular dimensions in PCAPass embeddings. As shown in Figure 4c, accuracy drops when the number of small singular values is high, particularly for $K \in \{2, 3, 4\}$. These results highlight the detrimental impact of rank deficiency on alignment accuracy.

We next investigate how singular value thresholding can mitigate rank deficiency in PCAPass. Specifically, we apply a threshold $\theta$, removing embedding dimensions with singular values $\Sigma_{i,i} \leq \theta \Sigma_{1,1}$, where $\Sigma_{1,1}$ is the largest singular value. We use $K = 10$ for undirected graphs and $K = 4$ for directed graphs, settings where PCAPass exhibits maximum rank deficiency.

The results are shown in Figure 6, with the threshold $\theta$ varying along the x-axis. We observe that selecting an appropriate $\theta$ effectively addresses rank deficiency, allowing PCAPass to achieve accuracies comparable to ACC. However, choosing $\theta$ requires careful consideration: it must be high enough to remove singular dimensions but not so high as to eliminate important information. The optimal range for $\theta$ depends on numerical precision and dataset characteristics; for Magna, only $\theta \in [10^{-6}, 10^{-5}]$ yields accuracies matching ACC.

Moreover, ACC consistently outperforms PCAPass in terms of computational speed by avoiding the computation of redundant features. As illustrated in Figures 6c and 6d, this speed advantage is substantial. On the Magna dataset, for instance, ACC is approximately 80% faster than PCAPass, with runtimes around 55 ms compared to 100 ms for PCAPass.

Finally, we compare ACC's graph alignment accuracy against three state-of-the-art graph alignment methods: S-GWL (Xu et al., 2019), CONE (Chen et al., 2020b), and FUGAL (Bommakanti et al., 2024). Here, ACC is significantly more efficient—approximately 100 times faster than CONE and FUGAL, and over 1,000 times faster than S-GWL. Additionally, ACC achieves the highest accuracy on the Enron dataset due to its effectiveness on dense, directed graphs, a scenario challenging for other methods. Detailed results of this comparison are provided in Appendix E.

## 5.2 Node classification: ACC vs self-supervised graph neural networks

To compare ACC with self-supervised graph neural networks (SSGNNs), we evaluate node classification accuracy on five standard directed graph datasets from the literature (Pei et al., 2020; Lim et al., 2021; Platonov et al., 2023; Rossi et al., 2023). Dataset details are provided in Appendix C.1. For initial node features $\boldsymbol{X}$, we use both the dataset-provided node features and the structural features used for graph alignment.

We train a gradient boosting classifier on top of the embeddings, following the setup of Sadowski et al. (2022). Specifically, we employ Scikit-learn's implementation of LightGBM (Ke et al., 2017) with default

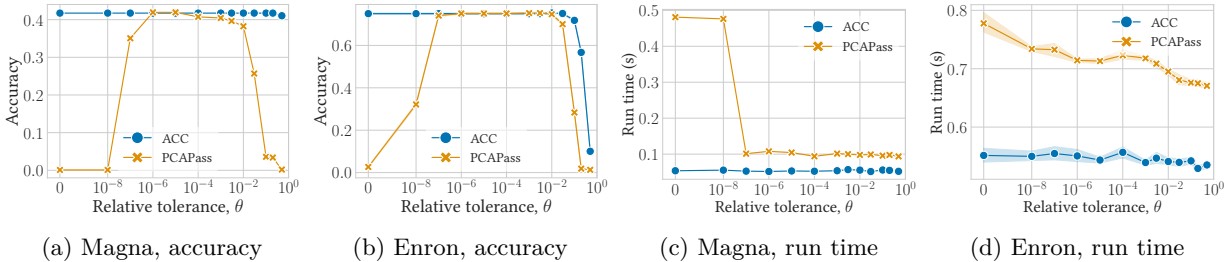

(a) Magna, accuracy     (b) Enron, accuracy     (c) Magna, run time     (d) Enron, run time

Figure 6: ACC and PCAPass for graph alignment with 15% noise edges, $K = 10$ for Magna, and $K = 4$ for Enron. The x-axes show the relative tolerance $\theta$ used to remove dimensions with small singular values. Figures 6a and 6b show accuracy on the y-axis, while 6c and 6d show run time. Markers and shaded areas indicate the average and standard deviation over 5 seeds. ACC achieves equal or superior accuracy for all $\theta$, and retains its accuracy for $\theta = 0$, as its embeddings are full rank. ACC is also consistently faster than PCAPass.

hyperparameters. To ensure robustness, we perform three repeats of 5-fold cross-validation with five different random seeds, reporting mean and standard deviation statistics for the classification accuracy.

All models are executed in a Google Cloud g2-standard-32 environment with one Nvidia L4 24GB GPU, 32 vCPUs @ 2.20GHz, and 128 GB of memory. Reported run times are averages over the five random seeds.

We adapt all baseline models to use directed message-passing following the approach outlined by Rossi et al. (2023), except for MVGRL (Hassani & Khasahmadi, 2020) and GREET (Liu et al., 2023), which feature non-standard architectures. Additionally, we introduce BGRL-GS and GraphMAEv2-GS by replacing the default GNNs in BGRL (Thakoor et al., 2022) and GraphMAEv2 (Hou et al., 2023) with the GraphSAGE model (Hamilton et al., 2017). SGCN and PCAPass are also included in our comparisons. For all models, including ACC, we use $K = 2$ message-passing iterations and $p = 512$ embedding dimensions, both of which are commonly used values in the literature (Hamilton et al., 2017; Veličković et al., 2018; Zhang et al., 2021; Thakoor et al., 2022; Hou et al., 2022). We use default values for optimizer and loss function hyperparameters. In the Appendix H, we provide a discussion on hyperparameter tuning of unsupervised learning models, as well as an analysis of huperparameter tuning results for the number of training epochs and embedding dimmensions for the SSGNNs.

Table 1 displays the accuracies and run times for all models and datasets. ACC stands out with the highest accuracy across all datasets. Moreover, these results demonstrate ACC's superior scalability compared to SSGNNs. On the Arxiv Year dataset, with 1 milion edges, ACC is over 70 times faster than CCA-SSG, the fastest SSGNN. Moreover, whereas most SSGNNs take a full day or more to run on Snap Patents, the largest dataset with 14 million edges, ACC requires *only 27 seconds.*

The linear model SGCN is the only baseline faster than ACC. However, SGCN suffers from over-smoothing, leading to ACC's substantial accuracy advantage on the Roman Empire dataset. This underscores how ACC's concatenation approach mitigates over-smoothing. Further results are available in the Appendix D.

Additionally, we observe a significant accuracy discrepancy between ACC and PCAPass on the Squirrel dataset, which cannot be attributed to the PCAPass rank deficiency. Our detailed analysis, available in Appendix F, highlights another benefit of ACC's message aggregation. Unlike PCAPass, which compresses its full embeddings in each iteration and may discard informative but low-variance features, ACC compresses features separately at each iteration, preserving information more evenly. Consequently, the ACC embeddings retains more information than PCAPass from the class-informative but low-variance features provided by $A_B X$ in the Squirrel dataset, resulting in higher accuracy.

# 6 Rank deficiency in SSGNNs: Beyond the linear setting

As demonstrated by our theoretical analysis in Section 3 and experimental results in Section 5, the embedding concatenation approach of PCAPass leads to rank deficiency, negatively impacting embedding quality. A natural question is whether similar issues affect self-supervised graph neural networks (SSGNNs), and if so, whether ACC's message aggregation approach could mitigate them.

Table 1: Gradient boosting node classification results. The top 3 accuracies are highlighted in bold. Snap Patents results were gathered using CPU only due to GPU memory limits. OOM abbreviates *out of memory*.

| Model | Chameleon Accuracy | Time | Squirrel Accuracy | Time | Roman Empire Accuracy | Time | Arxiv Year Accuracy | Time | Snap-Patents Accuracy | Time |
|---|---|---|---|---|---|---|---|---|---|---|
| No model, $\boldsymbol{X}$ | $64.6 \pm 2.3$ | 0ms | $52.1 \pm 1.4$ | 0ms | $70.4 \pm 0.7$ | 0ms | $44.7 \pm 0.2$ | 0ms | $56.1 \pm 0.1$ | 0ms |
| GAE[1] | $62.2 \pm 2.2$ | 11s | $44.8 \pm 1.5$ | 1m 4s | $53.4 \pm 3.1$ | 16s | $42.3 \pm 0.3$ | 7m 34s | $51.9 \pm 0.1$ | 3h 15m |
| DGI[2] | $61.5 \pm 2.3$ | 14s | $44.2 \pm 1.6$ | 1m 40s | $73.4 \pm 0.9$ | 4m 2s | $\boldsymbol{47.1 \pm 0.2}$ | 5h | Timeout | $\geq$24h |
| MVGRL[3] | $66.3 \pm 2.2$ | 26m 44s | $46.4 \pm 1.5$ | 26m 32s | $64.3 \pm 1.0$ | 39m 13s | OOM | $\geq$128 GB | OOM | $\geq$128 GB |
| BGRL[4] | $66.1 \pm 2.2$ | 7m 49s | $46.7 \pm 1.3$ | 31m 45s | $76.0 \pm 0.8$ | 25m 24s | $46.6 \pm 0.3$ | 4h 52m | Timeout | $\geq$24h |
| BGRL-GS[4] | $65.3 \pm 2.2$ | 8m 31s | $44.8 \pm 1.5$ | 33m 21s | $74.4 \pm 0.7$ | 21m 52s | $43.7 \pm 0.2$ | 3h 8m | Timeout | $\geq$24h |
| CCA-SSG[5] | $71.0 \pm 2.1$ | 4s | $\boldsymbol{59.9 \pm 1.2}$ | 7s | $63.7 \pm 0.7$ | 10s | $\boldsymbol{48.4 \pm 0.2}$ | 1m 11s | $\boldsymbol{56.1 \pm 0.0}$ | 1h 40m |
| GraphMAE[6] | $68.9 \pm 2.0$ | 27s | $56.9 \pm 1.8$ | 59s | $53.9 \pm 1.0$ | 1m 8s | $44.2 \pm 0.3$ | 8m 55s | $45.6 \pm 0.1$ | 16h |
| GraphMAEv2[7] | $69.6 \pm 1.9$ | 36s | $51.2 \pm 2.5$ | 1m 14s | $53.5 \pm 0.9$ | 1m 46s | $44.3 \pm 0.3$ | 14m 53s | $44.0 \pm 0.0$ | 23h |
| GraphMAEv2-GS[7] | $\boldsymbol{74.1 \pm 2.3}$ | 29s | $57.0 \pm 1.4$ | 1m 6s | $\boldsymbol{80.0 \pm 0.7}$ | 1m 35s | $46.3 \pm 0.3$ | 10m 46s | $53.8 \pm 0.1$ | 18h |
| GREET[8] | $61.4 \pm 2.2$ | 1m 25s | $44.2 \pm 1.6$ | 6m 45s | $\boldsymbol{80.1 \pm 0.5}$ | 1h 56m | OOM | $\geq$128 GB | OOM | $\geq$128 GB |
| SPGCL[9] | $66.3 \pm 2.2$ | 16s | $45.8 \pm 1.5$ | 1m 23s | $73.6 \pm 0.8$ | 55s | $46.4 \pm 0.2$ | 52m 6s | OOM | $\geq$128 GB |
| SGCN[10] | $\boldsymbol{74.7 \pm 1.8}$ | 374ms | $\boldsymbol{68.3 \pm 1.4}$ | 723ms | $45.4 \pm 0.8$ | 393ms | $45.0 \pm 0.2$ | 1s | $50.2 \pm 0.1$ | 26s |
| PCAPass[11] | $71.7 \pm 2.2$ | 2s | $51.5 \pm 1.3$ | 22s | $73.0 \pm 0.7$ | 852ms | $46.3 \pm 0.2$ | 2s | $\boldsymbol{61.2 \pm 0.1}$ | 2m 20s |
| ACC | $\boldsymbol{76.6 \pm 1.9}$ | 1s | $\boldsymbol{71.5 \pm 1.3}$ | 1s | $\boldsymbol{81.5 \pm 0.6}$ | 432ms | $\boldsymbol{49.4 \pm 0.3}$ | 1s | $\boldsymbol{62.6 \pm 0.1}$ | 27s |

[1] Kipf & Welling (2017)   [2] Veličković et al. (2019)   [3] Hassani & Khasahmadi (2020)   [4] Thakoor et al. (2022)
[5] Zhang et al. (2021)   [6] Hou et al. (2022)   [7] Hou et al. (2023)   [8] Liu et al. (2023)   [9] Wang et al. (2023)
[10] Wu et al. (2019)   [11] Sadowski et al. (2022)

The two main differences between SSGNNs and the PCAPass/ACC models are:

1. GNNs employ non-linear activation functions during message-passing.

2. Self-supervised GNNs require optimization of a loss function using non-convex methods due to these non-linearities.

These differences introduce challenges for both theoretical and empirical analyses. First, the non-linearities in GNNs invalidate the rank-deficiency analysis presented in Section 3. Consequently, more sophisticated mathematical frameworks are necessary, similar to ongoing theoretical investigations in deep learning (Roberts et al., 2022). Such advanced analyses are beyond the scope of this paper and constitute important directions for future research.

Empirical analysis is complicated by the diversity of existing SSGNN architectures, which differ significantly in terms of message-passing schemes (e.g., GCNs (Kipf & Welling, 2017) add self-loops, whereas GATs (Veličković et al., 2018) utilize attention mechanisms) and loss functions (Veličković et al., 2019; Zhang et al., 2021; Thakoor et al., 2022; Hou et al., 2022). Additionally, SSGNNs contain many hyperparameters influencing embedding quality, related to both architecture and optimization.

A comprehensive empirical study of rank deficiency and embedding quality across various SSGNN setups would therefore be extensive and is beyond this paper's scope. Nonetheless, we perform a small-scale analysis of SSGNN versions of PCAPass and ACC in Appendix G, summarized here for completeness.

Specifically, we convert PCAPass into an SSGNN model similar to GraphSAGE (Hamilton et al., 2017) by using the message-passing update:

$$\boldsymbol{H}^{(k)} = \sigma \left( \begin{bmatrix} \boldsymbol{H}^{(k-1)} & \boldsymbol{M}_{\mathrm{F}}^{(k)} & \boldsymbol{M}_{\mathrm{B}}^{(k)} \end{bmatrix} \boldsymbol{W}^{(k)} \right), \tag{12}$$

where $\sigma$ is the PReLU activation function (He et al., 2015), $\boldsymbol{W}^{(k)} \in \mathbb{R}^{3p_{k-1} \times p_k}$ are learnable weight matrices, and dimensions are set as $p_0 = d$ and $p_k = p_{\max}$ for $k \geq 1$. The message matrices $\boldsymbol{M}_{\mathrm{F}}^{(k)}$ and $\boldsymbol{M}_{\mathrm{B}}^{(k)}$ are computed as in Section 2. We refer to this model as PCAPass-GNN.

For the corresponding ACC-GNN model, we modify lines 3 and 6–7 in Algorithm 1 as follows:

$$\boldsymbol{M}^{(0)} = \sigma \left( \boldsymbol{X} \boldsymbol{W}^{(0)} \right), \qquad\qquad \boldsymbol{M}^{(k)} = \sigma \left( \begin{bmatrix} \boldsymbol{A}_{\mathrm{F}} \boldsymbol{M}^{(k-1)} & \boldsymbol{A}_{\mathrm{B}} \boldsymbol{M}^{(k-1)} \end{bmatrix} \boldsymbol{W}^{(k)} \right), \tag{13}$$

where $\boldsymbol{W}^{(0)} \in \mathbb{R}^{d \times c}$ and $\boldsymbol{W}^{(k)} \in \mathbb{R}^{2c \times c}$ for $k \geq 1$, with $c = p_{\max}/(K+1)$, consistent with standard ACC.

We train PCAPass-GNN and ACC-GNN using the GraphMAEv2 loss function (Hou et al., 2023), evaluating performance at 0, 20, and 200 training epochs to observe changes induced by weight learning. We assess both the embedding rank and the graph alignment accuracy across varying numbers of message-passing iterations, using a fixed embedding dimensionality of $p_{\max} = 512$. Full results are provided in Appendix G; key insights are summarized below.

Most notably, we find that PCAPass-GNN embeddings do not exhibit rank deficiency, as evidenced by the absence of significant gaps in their singular value spectra across all training epochs. This suggests that the inclusion of activation functions mitigates the explicit rank collapse observed in the linear setting. In terms of downstream performance, PCAPass-GNN shows substantial improvement over the original linear PCAPass model in graph alignment tasks and even surpasses ACC on one dataset.

Despite this improvement, two critical issues remain. First, PCAPass-GNN's alignment accuracy peaks around $K = 5$, after which performance declines as $K$ increases, suggesting a progressive loss of discriminative information. This pattern mirrors the behaviour observed for linear PCAPass in Figure 5, and stands in contrast to ACC-GNN, which, like ACC, maintains stable or plateauing accuracy across larger values of $K$.

Second, we observe that PCAPass-GNN achieves its highest alignment accuracy without training (0 epochs), with performance deteriorating after 20 and 200 training epochs. This performance deterioration appear to correlate with variability in the singular value spectra. Although the embeddings are not rank-deficient, we find that the condition number, i.e., the ratio between the largest and smallest singular values, increases significantly in many cases where alignment accuracy drops. This effect is particularly pronounced for undirected graphs, after 200 epochs of training, and at high message-passing depths (e.g., $K = 10$).

In contrast, ACC-GNN does not exhibit such degradation, instead maintaining more stable condition numbers and accuracies across training epochs, albeit with higher variance compared to linear ACC. However, these trends are not entirely consistent across all datasets and models, warranting further investigation.

From this exploratory analysis, we conclude that while PCAPass-GNN does not suffer from explicit rank deficiency, its embedding quality and behaviour closely resemble those of the linear PCAPass model. Meanwhile, ACC-GNN continues to mirror the robustness of linear ACC. This consistency suggests a deeper underlying reason for the benefits of ACC's message aggregation strategy—one that manifests as rank deficiency in the linear case and as instability in the non-linear setting. We hypothesize that large fluctuations in singular values may be linked to the observed performance degradation in PCAPass-GNN. As previously noted, formal theoretical analysis is needed to rigorously understand these effects, and this remains an important direction for future research in unsupervised node embedding models.

## 7 Conclusion and future work

In this paper, we addressed the issue of rank-deficient node embeddings generated by linear message-passing models, which not only lead to computational inefficiencies but also risk degrading downstream task performance. To overcome this, we introduced ACC, a novel unsupervised node embedding model that leverages message aggregation to avoid redundant feature computations, thereby ensuring full-rank embeddings.

Beyond our analysis of linear models, we also provided initial insights into rank instability in self-supervised graph neural networks, pointing to the existence of deeper underlying mechanisms behind the observed deficiencies. These findings motivate further theoretical and empirical investigation.

A promising direction for future work is to explore whether the layerwise PCA-based compression schemes used in ACC and PCAPass can be reformulated as the optimization of a global loss function. Such a formulation could deepen our theoretical understanding of these models and extend to a broader class of unsupervised message-passing architectures, including self-supervised graph neural networks.

Moreover, ACC's message aggregation strategy inherently ties each embedding feature to a specific hop distance within the graph. Replacing PCA with a feature selection mechanism could further enhance interpretability by explicitly linking each embedding dimension to a single input feature. This extension could be particularly valuable for applications requiring transparency, such as anomaly detection.

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

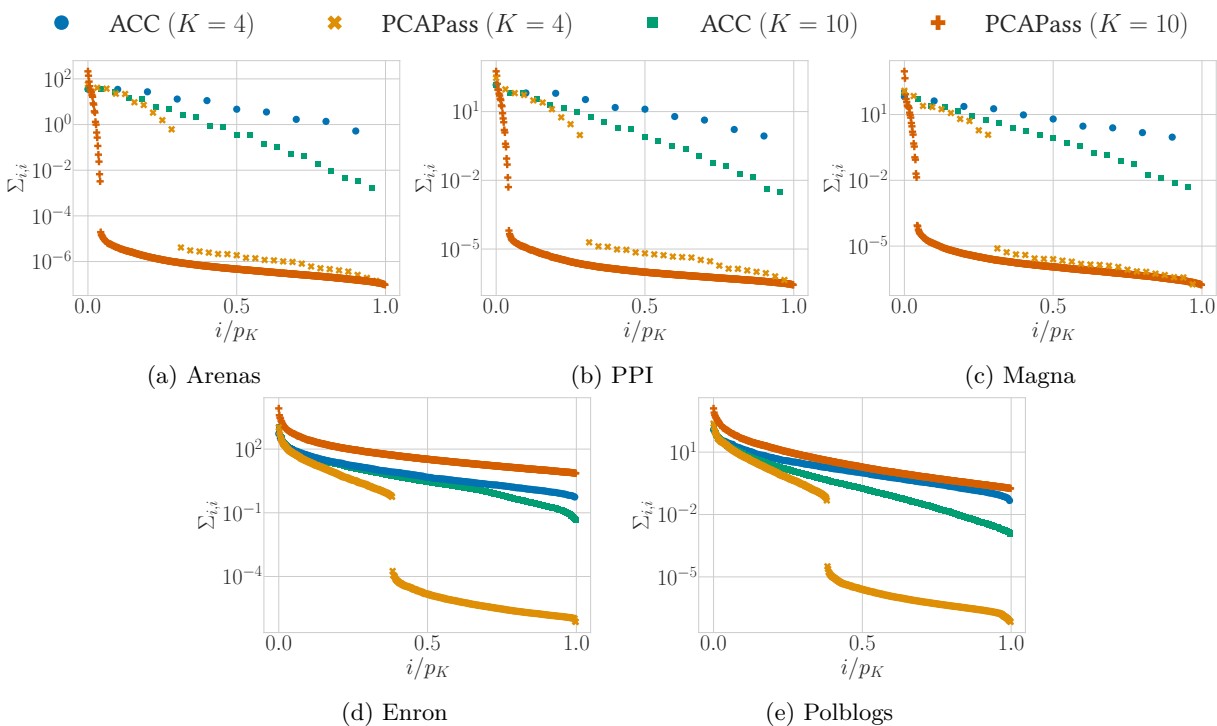

Figure 7: The singular values spectrum for ACC and PCAPass on all graph alignment datasets. Spectrums using both $K = 4$ and $K = 10$ message-passing iterations are shown. The y-axes show the singular values, and the x-axes their index in descending order, normalized using the number of embedding dimensions $p_K$.

## A  SVD of rank-deficient matrix

In Section 3.1, we stated the SVD of the matrix $\boldsymbol{Z} = \begin{bmatrix} \boldsymbol{X} & \boldsymbol{X} \end{bmatrix} \in \mathbb{R}^{n \times 2d}$, as $\boldsymbol{Z} = \boldsymbol{U_Z} \boldsymbol{\Sigma_Z} \boldsymbol{V_Z^\intercal}$, where

$$\boldsymbol{U_Z} = \begin{bmatrix} \boldsymbol{U} & \boldsymbol{\Upsilon_U} \end{bmatrix}, \qquad \boldsymbol{\Sigma_Z} = \begin{bmatrix} \sqrt{2}\boldsymbol{\Sigma} & \boldsymbol{0} \\ \boldsymbol{0} & \boldsymbol{0} \end{bmatrix}, \qquad \boldsymbol{V_Z^\intercal} = \frac{1}{\sqrt{2}} \begin{bmatrix} \boldsymbol{V^\intercal} & \boldsymbol{V^\intercal} \\ \boldsymbol{\Upsilon_{V_1}^\intercal} & \boldsymbol{\Upsilon_{V_2}^\intercal} \end{bmatrix}, \qquad (14)$$

and the block matrices $\boldsymbol{\Upsilon_U} \in \mathbb{R}^{n \times d}$, $\boldsymbol{\Upsilon_{V_1}} \in \mathbb{R}^{d \times d}$, and $\boldsymbol{\Upsilon_{V_2}} \in \mathbb{R}^{d \times d}$ each have orthogonal columns and satisfy the conditions $\boldsymbol{\Upsilon_U^\intercal} \boldsymbol{U} = \boldsymbol{0}$, $\boldsymbol{\Upsilon_U^\intercal} \boldsymbol{\Upsilon_U} = \boldsymbol{I}$, $\boldsymbol{\Upsilon_{V_1}^\intercal} \boldsymbol{V} = \boldsymbol{\Upsilon_{V_2}^\intercal} \boldsymbol{V} = \boldsymbol{0}$, and $\boldsymbol{\Upsilon_{V_1}^\intercal} \boldsymbol{\Upsilon_{V_1}} = \boldsymbol{\Upsilon_{V_2}^\intercal} \boldsymbol{\Upsilon_{V_2}} = \boldsymbol{I}$.

Here, we verify the orthonormality of $\boldsymbol{U_Z}$ and $\boldsymbol{V_Z}$ via matrix multiplication:

$$\boldsymbol{U_Z^\intercal} \boldsymbol{U_Z} = \begin{bmatrix} \boldsymbol{U^\intercal} \\ \boldsymbol{\Upsilon_U^\intercal} \end{bmatrix} \begin{bmatrix} \boldsymbol{U} & \boldsymbol{\Upsilon_U} \end{bmatrix} = \begin{bmatrix} \boldsymbol{U^\intercal U} & \boldsymbol{U^\intercal \Upsilon_U} \\ \boldsymbol{\Upsilon_U^\intercal U} & \boldsymbol{\Upsilon_U^\intercal \Upsilon_U} \end{bmatrix} = \begin{bmatrix} \boldsymbol{I} & \boldsymbol{0} \\ \boldsymbol{0} & \boldsymbol{I} \end{bmatrix}$$

$$\boldsymbol{V_Z^\intercal} \boldsymbol{V_Z} = \frac{1}{2} \begin{bmatrix} \boldsymbol{V^\intercal} & \boldsymbol{V^\intercal} \\ \boldsymbol{\Upsilon_{V_1}^\intercal} & \boldsymbol{\Upsilon_{V_2}^\intercal} \end{bmatrix} \begin{bmatrix} \boldsymbol{V} & \boldsymbol{\Upsilon_{V_1}} \\ \boldsymbol{V} & \boldsymbol{\Upsilon_{V_2}} \end{bmatrix} = \frac{1}{2} \begin{bmatrix} \boldsymbol{V^\intercal V + V^\intercal V} & \boldsymbol{V^\intercal \Upsilon_{V_1} + V^\intercal \Upsilon_{V_2}} \\ \boldsymbol{\Upsilon_{V_1}^\intercal V + \Upsilon_{V_2}^\intercal V} & \boldsymbol{\Upsilon_{V_1}^\intercal \Upsilon_{V_1} + \Upsilon_{V_2}^\intercal \Upsilon_{V_2}} \end{bmatrix} = \begin{bmatrix} \boldsymbol{I} & \boldsymbol{0} \\ \boldsymbol{0} & \boldsymbol{I} \end{bmatrix}.$$

## B  Singular values of ACC and PCAPass

Figure 7 presents the singular value spectra for the ACC and PCAPass embedding matrices across all graph alignment datasets. Both models use a maximum embedding dimension of $p_{\max} = 512$. However, this does not result in the same final embedding dimension, $p_K$, for each model. To facilitate comparison, we normalize the singular value index $i$ (x-axis) by the embedding dimension.

For the undirected graphs (Arenas, PPI, and Magna), the singular value gap indicating rank deficiency is clearly visible for PCAPass, with both $K = 4$ and $K = 10$ message-passing iterations. On the directed graphs, this gap only appears for $K = 4$, consistent with the results in Figure 4c. In contrast, ACC shows no such singular value gaps, further confirming that it produces full-rank embeddings.

Table 2: Table 2a shows graph statistics for the graph alignment datasets. Specifically, it shows the number of nodes and edges, the number of weakly and strongly connected components, the global clustering coefficient, $C_{\mathcal{G}}$, and the average path length, $\langle l_{\text{path}} \rangle$. Table 2b shows basic information regarding the node classification datasets: the number of nodes, edges and node features, and the number of node classes.

(a) Graph alignment datasets and statistics.

| Dataset | $n$ | $m$ | Dir. | # CC | # SCC | $C_{\mathcal{G}}$ | $\langle l_{\text{path}} \rangle$ |
|---|---|---|---|---|---|---|---|
| Arenas | 1.1K | 11K | ✗ | 1 | – | 0.17 | 3.6 |
| PPI | 3.9K | 76K | ✗ | 35 | – | 0.09 | 3.1 |
| Polblogs | 1.5K | 19K | ✓ | 268 | 688 | 0.25 | 3.4 |
| Enron | 7.9K | 142K | ✓ | 58 | 861 | 0.16 | 3.5 |
| Magna | 1K | 17K | ✗ | 1 | – | 0.62 | 5.5 |

(b) Node classification datasets.

| Dataset | $n$ | $m$ | # Feat. | # Cls. |
|---|---|---|---|---|
| Chameleon | 2.3K | 36K | 2325 | 5 |
| Squirrel | 5.2K | 217K | 2089 | 5 |
| Roman Empire | 23K | 33K | 300 | 18 |
| Arxiv Year | 169K | 1.2M | 128 | 5 |
| Snap patents | 2.9M | 14M | 269 | 5 |

## C   Additional experiments information and results

### C.1   Datasets

Table 2a provides statistics for the graph alignment datasets. Arenas (Guimera et al., 2003) is an undirected email network, where each edge represents email communication between two students. Similar to Magna, PPI (Breitkreutz et al., 2007) is a protein-protein interaction graph, with nodes representing proteins and edges denoting interactions between them. Polblogs (Adamic & Glance, 2005) is a hyperlink graph of political blogs, while Enron (Klimt & Yang, 2004) is an email communication network, where each node corresponds to an email address. Specifically, we use a subgraph of the full Enron dataset for our experiments.

Table 2b summarizes the node classification datasets. These datasets were used in recent work by Rossi et al. (2023) on directed message-passing for supervised graph neural networks.

The Chameleon and Squirrel datasets are both hyperlink networks, where each node represents a Wikipedia article, and edges indicate hyperlinks between articles. Node features are binary variables indicating the presence of specific nouns, while labels reflect the average monthly traffic for each webpage. Originally proposed by Rozemberczki et al. (2021) for regression tasks, they were later adapted for node classification by Pei et al. (2020).

The Roman Empire dataset, introduced by Platonov et al. (2023), is a word co-occurrence network based on the Wikipedia page for the Roman Empire. Nodes correspond to words, and edges represent syntactic dependencies between them, resulting in a graph that closely resembles a chain structure. The node labels represent syntactic roles, while the features are word embeddings.

Arxiv Year and Snap Patents were introduced by Lim et al. (2021) to benchmark GNNs on large-scale graphs. The Arxiv Year dataset is derived from the OGB Arxiv citation network (Hu et al., 2020), where nodes represent papers and features are derived from their abstracts. Unlike OGB Arxiv, which uses subject areas for labels, Arxiv Year assigns labels based on the publication year.

Snap Patents is a patent citation network, where nodes represent patents and edges indicate citations. Originally studied by Leskovec et al. (2005) to investigate the evolution of citation networks over time, the dataset used by Lim et al. (2021) assigns labels based on the year each patent was granted. Node features are generated from patent metadata.

### C.2   Additional graph alignment results

Figure 8 extends the results shown in Figure 6 from the main paper, now including the Arenas, PPI, and Polblogs datasets. The figure presents graph alignment accuracies and run times across varying relative tolerances $\theta$ for thresholding singular values.

The trends in accuracy and run time closely resemble those observed for Magna and Enron in Figure 6. Specifically, PCAPass can match ACC's accuracy within a certain range of $\theta$ values, but its accuracy sharply drops to zero outside this range. In contrast, ACC maintains stable accuracy, only declining for very

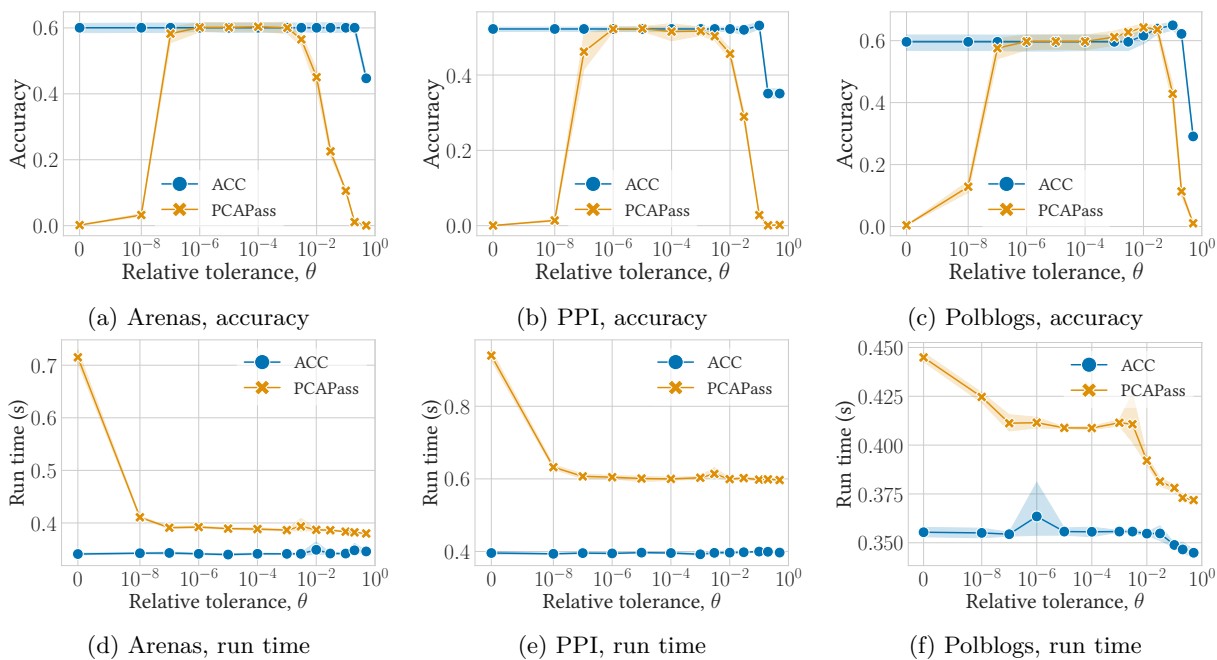

Figure 8: ACC and PCAPass for graph alignment with 15% noise edges, $K = 10$ for Arenas and PPI, and $K = 4$ for Polblogs. The x-axes show the relative tolerance $\theta$ applied to remove dimensions with small singular values. Figures 8a to 8c show accuracy on the y-axis, while 8d to 8f show run time. Markers and shaded areas indicate the average and standard deviation over 5 seeds.

high values of $\theta$, which leads to excessive removal of embedding features. Additionally, ACC consistently outperforms PCAPass in terms of run time.

### C.3 Node classification baselines

Below, we list the baselines used in our node classification experiment, including references to the respective model implementations and their licences. For models licensed under the MIT or Apache 2.0 licences, we also release our directed extensions as part of this paper's code repository. Additionally, we specify the default number of epochs used for training, as this directly influences the reported model run times. For other hyperparameter defaults, please refer to our code.

**GAE** (Kipf & Welling, 2017): `https://pytorch-geometric.readthedocs.io/en/latest/generated/torch_geometric.nn.models.GAE.html`, MIT Licence, 200 epochs.

**DGI** (Veličković et al., 2019): `https://github.com/PetarV-/DGI`, MIT licence, 100 epochs.

**MVGRL** (Hassani & Khasahmadi, 2020): `https://github.com/kavehhassani/mvgrl`, No licence, 3000 epochs.

**BGRL** (Thakoor et al., 2022): `https://github.com/nerdslab/bgrl`, Apache Licence 2.0, 10000 epochs.

**CCA-SSG** (Zhang et al., 2021): `https://github.com/hengruizhang98/CCA-SSG`, Apache Licence 2.0, 100 epochs.

**GraphMAE** (Hou et al., 2022): `https://github.com/THUDM/GraphMAE`, MIT licence, 1000 epochs.

**GraphMAEv2** (Hou et al., 2023): `https://github.com/THUDM/GraphMAE2`, MIT licence, 1000 epochs.

**GREET** (Liu et al., 2023): `https://github.com/yixinliu233/GREET`, MIT licence, 400 epochs.

**SPGCL** (Wang et al., 2023): `https://github.com/haonan3/SPGCL`, No licence, 500 epochs.

Table 3: Node classification results using a logistic regression classifier. OOM abbreviates *out of memory*. All Snap Patents results were gathered using CPU only as the SSGNNs exceeded our GPU memory limit. The top 3 accuracies for each dataset are highlighted in bold.

| Model | Chameleon | | Squirrel | | Roman Empire | | Arxiv Year | | Snap-Patents | |
|---|---|---|---|---|---|---|---|---|---|---|
| | Accuracy | Time | Accuracy | Time | Accuracy | Time | Accuracy | Time | Accuracy | Time |
| No model, $\mathbf{X}$ | $52.3 \pm 2.2$ | 0ms | $35.3 \pm 1.4$ | 0ms | $69.8 \pm 0.7$ | 0ms | $43.4 \pm 0.2$ | 0ms | $50.7 \pm 0.1$ | 0ms |
| GAE[1] | $54.8 \pm 2.6$ | 11s | $38.2 \pm 1.9$ | 1m 4s | $66.1 \pm 1.8$ | 16s | $43.2 \pm 0.3$ | 7m 34s | $49.3 \pm 0.1$ | 3h 15m |
| DGI[2] | $54.0 \pm 2.8$ | 14s | $40.4 \pm 1.7$ | 1m 40s | $76.5 \pm 0.7$ | 4m 2s | $\mathbf{49.4 \pm 0.3}$ | 5h | Timeout | $\geq$24h |
| MVGRL[3] | $55.4 \pm 3.0$ | 26m 44s | $40.0 \pm 1.4$ | 26m 32s | $65.1 \pm 0.9$ | 39m 13s | OOM | $\geq$128 GB | OOM | $\geq$128 GB |
| BGRL[4] | $55.9 \pm 2.6$ | 7m 49s | $42.8 \pm 1.5$ | 31m 45s | $\mathbf{79.0 \pm 0.7}$ | 25m 24s | $49.4 \pm 0.3$ | 4h 52m | Timeout | $\geq$24h |
| BGRL-GS[4] | $58.6 \pm 2.1$ | 8m 31s | $41.6 \pm 1.7$ | 33m 21s | $78.9 \pm 0.7$ | 21m 52s | $47.8 \pm 0.3$ | 3h 8m | Timeout | $\geq$24h |
| CCA-SSG[5] | $59.5 \pm 2.7$ | 4s | $41.6 \pm 1.4$ | 7s | $70.0 \pm 0.7$ | 10s | $\mathbf{50.9 \pm 0.3}$ | 1m 11s | $\mathbf{55.3 \pm 0.1}$ | 1h 40m |
| GraphMAE[6] | $\mathbf{65.3 \pm 2.1}$ | 27s | $\mathbf{43.5 \pm 1.7}$ | 59s | $55.7 \pm 0.8$ | 1m 8s | $43.8 \pm 0.3$ | 8m 55s | $44.1 \pm 0.1$ | 16h |
| GraphMAEv2[7] | $\mathbf{65.1 \pm 2.2}$ | 36s | $40.4 \pm 2.1$ | 1m 14s | $55.0 \pm 0.9$ | 1m 46s | $44.1 \pm 0.3$ | 14m 53s | $42.7 \pm 0.1$ | 23h |
| GraphMAEv2-GS[7] | $\mathbf{70.6 \pm 2.3}$ | 29s | $\mathbf{48.6 \pm 1.6}$ | 1m 6s | $\mathbf{80.2 \pm 0.7}$ | 1m 35s | $46.9 \pm 0.3$ | 10m 46s | $\mathbf{54.7 \pm 0.0}$ | 18h |
| GREET[8] | $53.9 \pm 2.3$ | 1m 25s | $37.1 \pm 1.5$ | 6m 45s | $77.5 \pm 0.6$ | 1h 56m | OOM | $\geq$128 GB | OOM | $\geq$128 GB |
| SPGCL[9] | $58.2 \pm 2.3$ | 16s | $42.5 \pm 1.5$ | 1m 23s | $76.2 \pm 0.7$ | 55s | $48.2 \pm 0.3$ | 52m 6s | OOM | $\geq$128 GB |
| SGCN[10] | $49.8 \pm 2.7$ | 374ms | $35.3 \pm 1.2$ | 723ms | $39.2 \pm 0.7$ | 393ms | $43.2 \pm 0.2$ | 1s | $42.3 \pm 0.7$ | 26s |
| PCAPass[11] | $48.7 \pm 2.1$ | 2s | $40.5 \pm 1.3$ | 22s | $77.6 \pm 0.7$ | 852ms | $49.3 \pm 0.3$ | 2s | $54.5 \pm 0.1$ | 2m 20s |
| ACC | $60.7 \pm 2.5$ | 1s | $\mathbf{44.1 \pm 1.4}$ | 1s | $\mathbf{79.3 \pm 0.6}$ | 432ms | $\mathbf{49.4 \pm 0.3}$ | 1s | $\mathbf{56.6 \pm 0.0}$ | 27s |

[1] Kipf & Welling (2017)  [2] Veličković et al. (2019)  [3] Hassani & Khasahmadi (2020)  [4] Thakoor et al. (2022)
[5] Zhang et al. (2021)  [6] Hou et al. (2022)  [7] Hou et al. (2023)  [8] Liu et al. (2023)  [9] Wang et al. (2023)
[10] Wu et al. (2019)  [11] Sadowski et al. (2022)

**SGCN** (Wu et al., 2019): `https://pytorch-geometric.readthedocs.io/en/latest/generated/torch_geometric.nn.conv.SGConv.html`, MIT licence.

**PCAPass** (Sadowski et al., 2022): The original PCAPass implementation is available at `https://github.com/krzysztof-daniell/PCAPass` under the MIT License. However, we use our reimplementation for this paper, available alongside ACC in our online code repository.

### C.4 Node classification results using logistic regression

Table 3 presents the node classification test accuracies for each embedding model and dataset using a *logistic regression* classifier. The experimental setup otherwise follows the description in Section 5.2. We observe that the accuracies obtained with logistic regression are generally lower than those achieved using the gradient boosting classifier in the main paper (Table 1). This is expected, as gradient boosting is a more expressive and less biased model capable of capturing highly non-linear class boundaries.

Overall, ACC embeddings perform competitively against the SSGNNs also using the logistic regression classifier, achieving the highest accuracy on Snap Patents, the second-highest on Arxiv Year and Roman Empire, and ranking third on Squirrel and fourth on Chameleon.

On the Chameleon and Squirrel datasets, we observe a notable drop in ACC's performance with logistic regression compared to gradient boosting: from 76.6% to 60.7% on Chameleon, and from 71.5% to 44.1% on Squirrel. A similar trend is seen for other linear embedding models like SGCN and PCAPass. This suggests that a non-linear classification model is necessary to fully exploit the information in these embeddings.

In contrast, the performance gap is smaller for several SSGNNs. For instance, GraphMAEv2-GS sees only a modest decline in accuracy from 74.1% to 70.6% on Chameleon. This indicates that the inherent non-linearity of GNNs can compensate for the simplicity of logistic regression. However, the extent to which this potential is realized depends on the specific GNN architecture and training, as many SSGNN models still achieve lower accuracies than ACC with logistic regression.

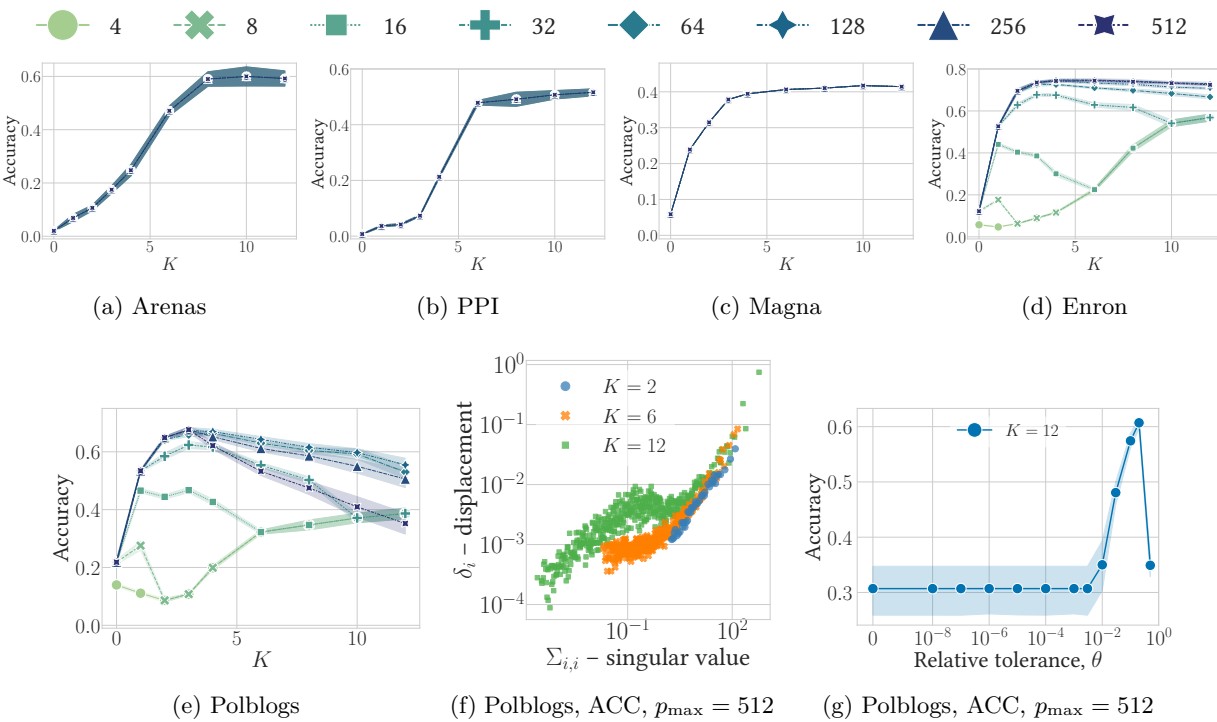

Figure 9: Figures 9a to 9e present the ACC graph alignment accuracy (y-axis) for varying numbers of message-passing iterations $K$ (x-axis) and embedding dimensions $p_{\max}$ (indicated by hue and line style). Figures 9f and 9g provide deeper insights into the results for the Polblogs dataset. Similar to Figure 3c, Figure 9f plots the singular values of the embedding matrix (x-axis) against the average displacement along the corresponding principal axis, $\delta_i$ (y-axis). For $K = 6$ and $K = 12$, the correlation between singular value and displacement observed at $K = 2$ is disrupted, with some features exhibiting larger displacements than expected based on their singular values. These features cause the accuracy drop shown in Figure 9e. Finally, Figure 9g confirms that by removing the dimensions corresponding to these high-displacement, low-singular-value features, the alignment accuracy can be restored.

## D  Effect of the number of message-passing iterations and embedding dimensions

In this section, we investigate how the quality of ACC embeddings is influenced by the two primary hyperparameters: the number of message-passing iterations, $K$, and the maximum embedding dimensionality, $p_{\max}$. To assess their impact, we replicate the graph alignment and node classification experiments, measuring the accuracy of ACC embeddings across a grid of $K$ and $p_{\max}$ values.

### D.1  Graph Alignment

Figures 9a to 9e present the grid evaluation results for graph alignment, where the x-axes denote the number of message-passing iterations, $K$. The hue and line style differentiate the $p_{\max}$ values.

The first key observation is that accuracy increases steadily with $K$ for each undirected graph, eventually plateauing. This illustrates ACC's ability to preserve information across multiple message-passing iterations.

Secondly, for the undirected graphs, the value of $p_{\max}$ appears to have no significant effect. This is because the initial number of features, $d = 2$, matches the minimum compression dimensionality, $c_{\min} = 2$, for these datasets. Therefore, regardless of the value of $p_{\max}$, the resulting ACC embeddings will have dimensionality $p_K = 2 \cdot (K + 1)$.

On the directed graphs, we observe different behaviour. Starting with Enron in Figure 9d, when $p_{\max}$ is sufficiently high, the accuracy follows a similar pattern to the undirected graphs, increasing steadily before levelling off. However, for $p_{\max} \in 8, 16, 32$, a different trend emerges: the accuracy initially increases, but then decreases, eventually aligning with the curve for $p_{\max} = 4$.

This seemingly unusual behaviour stems from the formula used to determine the number of compression dimensions, $c = \max\left(\lfloor p_{\max}/(K+1) \rfloor, c_{\min}\right)$, which in turn defines the final embedding dimensionality, $p_K = c \cdot (K+1)$. As $K$ increases, this formula can lead to a decrease in the final embedding dimensionality, $p$. Consequently, more information must be compressed into fewer dimensions, resulting in a loss of information and a corresponding drop in accuracy. However, once $K + 1$ becomes a factor of $p_{\max}$, the value of $p_K$ increases again, restoring some of the lost accuracy.

For example, with $p_{\max} = 8$, we obtain $p_0 = 6$ and $p_1 = 8$ for $K = 0$ and $K = 1$, but then $p_2 = 6$ for $K = 2$, before increasing again to $p_3 = 8$. Beyond $K \geq 4$, the formula sets $c = c_{\min} = 2$, which is why the curve for $p_{\max} = 8$ converges with the curve for $p_{\max} = 4$. A similar explanation applies to the curves for $p_{\max} \in 16, 32$, as shown in Figure 9d.

In Figure 9e, we observe the same effect for $p_{\max} \in \{8, 16, 32\}$ on Polblogs as we did for Enron. However, we also notice a distinctly different behaviour: as $K$ increases, the accuracy begins to drop for $p_{\max} \geq 64$.

This decline in accuracy is neither due to information loss from compression nor rank deficiency. As seen in Figure 7e, the singular value spectrum for ACC on Polblogs with $K = 10$ does not exhibit any singular value gaps. Instead, the drop in accuracy occurs because increasing $K$ generates embedding features that are disproportionately noisy.

We demonstrate this effect in Figure 9f. Similar to Figures 3c and 3d in the main paper, Figure 9f plots the singular values of the embedding matrix against the average displacement along the corresponding principal axis due to noise from graph alignment. As a reminded from the main paper, let $\boldsymbol{Z}^{(1)}$ represent the embedding matrix for graph $\mathcal{G}_1$, and $\boldsymbol{Z}^{(2)}$ the embedding matrix for the noisy graph $\mathcal{G}_2$. Using the singular value decomposition $\boldsymbol{U}, \boldsymbol{\Sigma}, \boldsymbol{V}^\intercal = \boldsymbol{Z}^{(1)}$, where $\boldsymbol{V}$ contains the principal axes for $\boldsymbol{Z}^{(1)}$, the x-axis shows the singular values $\Sigma_{i,i}$, and the y-axis shows the displacement $\delta_i = \frac{1}{n}\|(\boldsymbol{Z}^{(1)} - \boldsymbol{Z}^{(2)})\boldsymbol{V}_{:,i}\|_2$ along the $i$th principal axis.

For $K = 2$, we observe that the displacement $\delta_i$ is proportional to the singular values $\Sigma_{i,i}$, consistent with our observations across the other four graph alignment datasets. However, for $K = 6$ and $K = 10$, this linear correlation breaks down, and features with disproportionately large displacements emerge. This is seen as the curvature of the point clouds in Figure 9f.

We further verify that these high-displacement features cause the accuracy drop by applying singular value thresholding. The results for $K = 12$ and $p_{\max} = 512$ are shown in Figure 9g. As can be seen, the alignment accuracy improves dramatically, from 30% to 60%, once the dimensions with small singular values and high displacements are removed.

We do not yet fully understand why the high-displacement dimensions appear in the Polblogs embeddings. Looking at the graph statistics in Table 2a, two potential causes stand out: the large number of weakly connected components and the high global clustering coefficient for Polblogs. However, we can rule out the former, as we observe the same behaviour when running the graph alignment experiment on the largest connected component of Polblogs. This leaves the high global clustering coefficient as the most likely cause. Further research is needed to verify this hypothesis and to explore the underlying mechanisms that might lead to the emergence of these high-displacement dimensions.

## D.2  Node classification

Figure 10 illustrates the effect of $K$ and $p_{\max}$ on ACC node classification accuracies, measured using a logistic regression classifier. Regarding $p_{\max}$, we observe a consistent increase in accuracy across all values of $K$. This is expected, as higher embedding dimensionality allows the embeddings to capture more information, facilitating better classification performance.

The effect of increasing the number of message-passing iterations $K$ depends on the embedding dimensionality $p_{\max}$. When $p_{\max}$ is sufficiently large, classification accuracy rises and eventually plateaus for all four datasets.

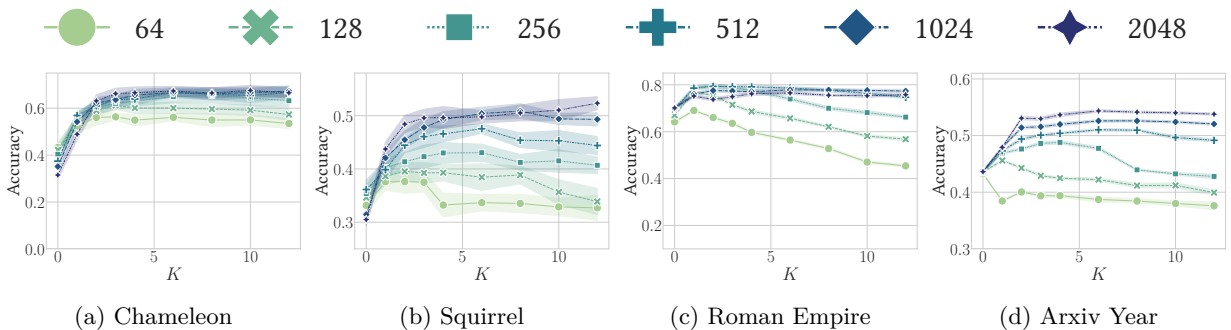

(a) Chameleon  (b) Squirrel  (c) Roman Empire  (d) Arxiv Year

Figure 10: These figures the ACC node classification accuracy (y-axis) for various number of message-passing iterations $K$ (x-axis), and embedding dimensions $p_{\max}$ (hue and style).

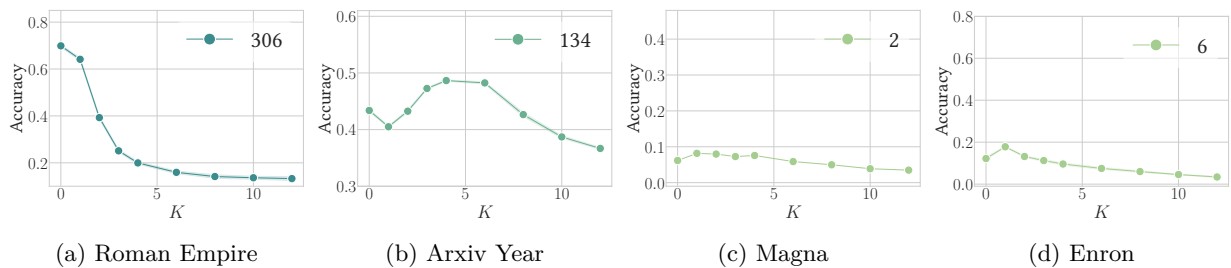

(a) Roman Empire  (b) Arxiv Year  (c) Magna  (d) Enron

Figure 11: These figures show the SGCN (Wu et al., 2019) node classification accuracy (Figures 11a and 11b) and graph alignment accuracy (Figures 11c and 11d) on the y-axis, plotted against the number of message-passing iterations $K$ (x-axis). The legends indicate the number of SGCN embedding dimensions, which are always equal to the number of input features $d$.

However, when $p_{\max}$ is too small, accuracy can decline as $K$ increases. This effect is particularly notable for the Roman Empire dataset, as seen in Figure 10c.

The drop in accuracy is due to the increased need for compression as $K$ grows, meaning that less information from each scale of the graph is retained. Specifically, when $K = 0$, the initial $d$ features are compressed into $p_{\max}$ embedding dimensions, but when $K = 12$, these features are compressed into only $\lfloor p_{\max}/13 \rfloor$ dimensions. The loss of information due to this increased compression results in an accuracy decline for datasets where local features (i.e., small $K$) are especially important for classification. The Roman Empire dataset exemplifies this behaviour, where the information contained in $\boldsymbol{X}$, $\boldsymbol{A}_{\mathrm{F}}\boldsymbol{X}$, and $\boldsymbol{A}_{\mathrm{B}}\boldsymbol{X}$ is critical for achieving high accuracy.

### D.3 Over-smoothing in SGCN

To highlight the advantage of the concatenation update used by ACC, we provide results using SGCN (Wu et al., 2019) in Figure 11. SGCN employs summation rather than concatenation to update its embeddings. This results in a loss of information with each message-passing iteration, leading to over-smoothing (Li et al., 2018; Chen et al., 2020a).

This issue is particularly noticeable on the Roman Empire dataset, where the accuracy of SGCN drops from 70% to 10% as $K$ increases. In contrast, ACC's accuracy remains stable as long as a sufficiently high embedding dimension is used, as shown in Figure 10c.

Additionally, SGCN embeddings maintain a fixed dimension for all message-passing iterations, $p_K = d$. This limitation hampers graph alignment accuracy, as it prevents the integration of information from different scales to form more distinct embeddings. Consequently, the alignment accuracy for SGCN is lower, as evidenced in Figures 11c and 11d.

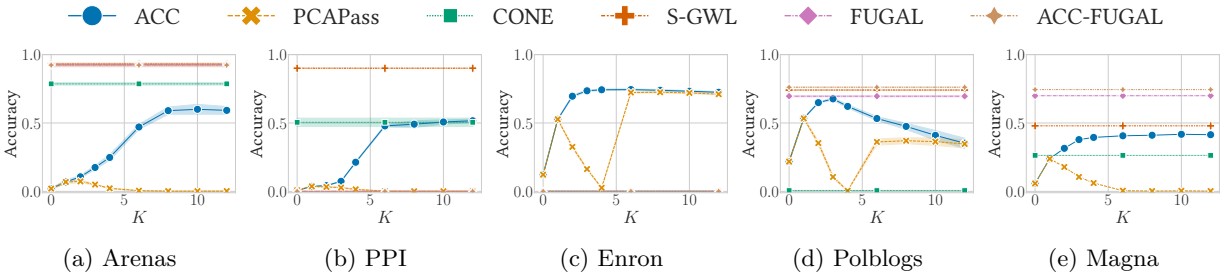

Figure 12: Graph alignment accuracy comparison between ACC, PCAPass, and graph alignment baselines under 15% edge noise. The x-axis denotes the number of message-passing iterations used for ACC and PCAPass. For each method (CONE, S-GWL, FUGAL, ACC-FUGAL), markers and shaded regions indicate the mean and standard deviation over three random seeds. The results for ACC and PCAPass are repeated from Figure 5, where five were used.

# E  Comparison to state-of-the-art graph alignment methods

In this section, we compare the graph alignment accuracy of ACC against three state-of-the-art graph alignment algorithms: S-GWL (Xu et al., 2019), CONE (Chen et al., 2020b), and FUGAL (Bommakanti et al., 2024). None of these methods natively support directed graphs, which is a limitation shared by most recent graph alignment algorithms (Skitsas et al., 2023). Fortunately, S-GWL naturally generalizes to directed graphs, and FUGAL can be extended in a straightforward manner since it is node feature-based. Specifically, we compute node features on both the original and transposed graphs, and concatenate the resulting feature vectors as input to FUGAL's optimization protocol. This simple modification enables FUGAL to process directed graphs. In contrast, adapting CONE to directed graphs is non-trivial due to its architecture. For all methods, we adopt the recommended hyperparameter settings from Skitsas et al. (2023).

Figure 12 shows the previously reported graph alignment accuracy results for ACC and PCAPass (with whitening), now presented alongside the newly added results for each baseline method. The x-axis represents the number of message-passing iterations $K$ used for ACC and PCAPass. The corresponding run times for all methods are provided in Figure 13.

We begin our comparison by examining ACC alongside CONE, which is the weakest of the three baseline methods. As expected, CONE fails entirely on the directed graphs Enron and Polblogs, reflecting its lack of support for directed edges. On the undirected datasets, CONE and ACC achieve similar levels of accuracy: CONE slightly outperforms ACC on Arenas, ACC performs better on Magna, and the two models yield identical results on PPI. However, when it comes to computational efficiency, ACC holds a clear advantage. While CONE exhibits quadratic time complexity in the number of nodes, $\mathcal{O}(n^2)$, ACC operates with linear complexity, as discussed in Section 4.1. In practice, this difference is substantial and ACC is approximately 100 times faster than CONE, as shown in Figure 13.

S-GWL is the strongest overall baseline in terms of alignment accuracy, achieving the highest performance on Arenas, PPI, and Polblogs. This result is consistent with the benchmark evaluation by Skitsas et al. (2023), where S-GWL was identified as the top-performing method.

However, S-GWL is also by far the most computationally expensive of the three baselines, as shown in Figure 13. On the Enron dataset, S-GWL failed to converge even after two hours of run time, at which point we terminated its execution. In contrast, ACC is highly efficient; approximately 1000 times faster across the five datasets. This efficiency stems from two key factors. First, S-GWL has a time complexity of $\mathcal{O}(n^2 \log n)$ (Skitsas et al., 2023), while ACC scales linearly with the number of nodes. Second, S-GWL is an iterative optimization-based method, whereas ACC generates embeddings in a single forward pass.

FUGAL is the second-best baseline in terms of alignment accuracy, achieving the highest performance on Magna and tying with S-GWL for best results on Arenas. Notably, our extension of FUGAL to directed graphs allows it to perform well on Polblogs, where its accuracy closely matches that of both S-GWL and

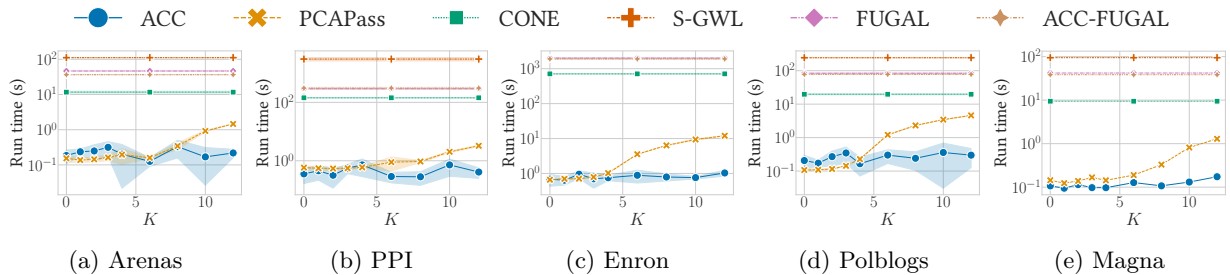

(a) Arenas  (b) PPI  (c) Enron  (d) Polblogs  (e) Magna

Figure 13: Run time comparison of ACC and PCAPass to graph alignment baselines. The x-axis shows the number of message-passing iterations $K$ and the y-axis the algorithm run time in seconds. Markers and shaded areas represent the average and standard deviation over three seeds for the graph alignment baselines, and five seeds for ACC and PCAPass.

ACC. However, FUGAL fails completely on PPI and Enron, achieving only trivial alignment accuracy. In contrast, ACC maintains high accuracy on both datasets while also being significantly more efficient, as FUGAL has cubic time complexity, $\mathcal{O}(n^3)$.

To understand why FUGAL fails on PPI and Enron, we examine the graph statistics in Table 2a. These two graphs are simultaneously the densest in the benchmark and exhibit the lowest global clustering coefficients. While Bommakanti et al. (2024) showed that FUGAL performs well across varying graph densities, their experiments were conducted in noise-free settings. We therefore hypothesize that the combination of edge-removal noise and high density poses challenges for FUGAL.

This limitation could stem from either FUGAL's initial feature extraction step or its subsequent optimization procedure. To investigate, we replace FUGAL's node features with ACC embeddings computed using $K = 6$ and $p_{\max} = 512$. The resulting method, denoted ACC-FUGAL, is shown in Figures 12 and 13. As illustrated, ACC-FUGAL also fails on PPI and Enron, indicating that the bottleneck lies in FUGAL's optimization stage rather than its input features.

On the other three datasets, however, ACC-FUGAL achieves the highest accuracy among all methods. This suggests that, if FUGAL's limitations on dense graphs can be addressed, combining ACC embeddings with the FUGAL optimization framework could yield a highly effective alignment algorithm. In the meantime, using ACC on its own offers a compelling alternative as it delivers strong accuracy with far greater efficiency, and scaling well to large graphs due to its linear time complexity.

## F    Analysis of node classification accuracy on the Squirrel dataset

In our node classification benchmark results using the gradient boosting classifier, as shown in Table 1, ACC achieves an accuracy of 72% on the Squirrel dataset, whereas PCAPass achieves only 52%. In this section, we explore the source of this discrepancy by analysing the Squirrel dataset and comparing the features generated by message aggregation and embedding aggregation.

To identify the key features contributing to high classification accuracy on Squirrel, we perform a single message-passing iteration to obtain three feature matrices: $\boldsymbol{X} \in \mathbb{R}^{n \times d}$, $\boldsymbol{A}_{\mathsf{F}}\boldsymbol{X} \in \mathbb{R}^{n \times d}$, and $\boldsymbol{A}_{\mathsf{B}}\boldsymbol{X} \in \mathbb{R}^{n \times d}$. We refer to these as *feature groups*.

By training a gradient boosting classifier on each feature group with an 80-20 training-test split, we find the following test accuracies: 51% for $\boldsymbol{X}$, 33% for $\boldsymbol{A}_{\mathsf{F}}\boldsymbol{X}$, and 84% for $\boldsymbol{A}_{\mathsf{B}}\boldsymbol{X}$. These results indicate that the features in $\boldsymbol{A}_{\mathsf{B}}\boldsymbol{X}$ are particularly crucial for achieving high classification accuracy on the Squirrel dataset.

Next, we investigate how well each feature group is preserved in the PCAPass and ACC embeddings. Ignoring the column centring step of PCA, we can express the PCAPass embeddings after one message-passing

iteration as

$$\boldsymbol{Z} = \boldsymbol{H}^{(1)} = [\boldsymbol{X}, \boldsymbol{A}_\mathrm{F}\boldsymbol{X}, \boldsymbol{A}_\mathrm{B}\boldsymbol{X}]\boldsymbol{W} = \boldsymbol{X}\boldsymbol{W}_X + \boldsymbol{A}_\mathrm{F}\boldsymbol{X}\boldsymbol{W}_\mathrm{F} + \boldsymbol{A}_\mathrm{B}\boldsymbol{X}\boldsymbol{W}_\mathrm{B}, \qquad \boldsymbol{W} = \begin{bmatrix} \boldsymbol{W}_X \\ \boldsymbol{W}_\mathrm{F} \\ \boldsymbol{W}_\mathrm{B} \end{bmatrix}, \qquad (15)$$

where $\boldsymbol{W} \in \mathbb{R}^{3d\times p}$ is the projection matrix learned via PCA. We divide this matrix vertically into three sub-matrices: $\boldsymbol{W}_X \in \mathbb{R}^{d\times p}$, $\boldsymbol{W}_\mathrm{F} \in \mathbb{R}^{d\times p}$, and $\boldsymbol{W}_\mathrm{B} \in \mathbb{R}^{d\times p}$. These matrices compress $\boldsymbol{X}$, $\boldsymbol{A}_\mathrm{F}\boldsymbol{X}$, and $\boldsymbol{A}_\mathrm{B}\boldsymbol{X}$ respectively. By analysing these matrices, we can assess how much information from each feature group is preserved.

Specifically, for each of the $p = 512$ features, we compute the proportion of each feature group that contributes to the embedding. Since each column in $\boldsymbol{W}$ has unit norm, these proportions can be calculated as follows:

$$\mathbf{w}_X = \sum_{i=1}^{d} (\boldsymbol{W}_X)_{i,:}^2 , \qquad \mathbf{w}_\mathrm{F} = \sum_{i=1}^{d} (\boldsymbol{W}_\mathrm{F})_{i,:}^2 , \qquad \mathbf{w}_\mathrm{B} = \sum_{i=1}^{d} (\boldsymbol{W}_\mathrm{B})_{i,:}^2 , \qquad (16)$$

where $\mathbf{w}_X$, $\mathbf{w}_\mathrm{F}$, and $\mathbf{w}_\mathrm{B}$ denote the proportions of each feature group represented in the embeddings. Note that $\mathbf{w}_X + \mathbf{w}_\mathrm{F} + \mathbf{w}_\mathrm{B} = \mathbf{1}_p$, where $\mathbf{1}_p$ is a length-$p$ vector of ones.

We can perform a similar analysis for ACC. In this case, the embeddings are given by $\boldsymbol{Z} = [\boldsymbol{M}^{(0)}, \boldsymbol{M}^{(1)}]$, where $\boldsymbol{M}^{(0)} = \boldsymbol{X}\boldsymbol{V}_X$ and

$$\boldsymbol{M}^{(1)} = [\boldsymbol{A}_\mathrm{F}\boldsymbol{M}^{(0)}, \boldsymbol{A}_\mathrm{B}\boldsymbol{M}^{(0)}]\boldsymbol{V} = \boldsymbol{A}_\mathrm{F}\boldsymbol{X}\boldsymbol{V}_X\boldsymbol{V}_\mathrm{F} + \boldsymbol{A}_\mathrm{B}\boldsymbol{X}\boldsymbol{V}_X\boldsymbol{V}_\mathrm{B}, \qquad \boldsymbol{V} = \begin{bmatrix} \boldsymbol{V}_\mathrm{F} \\ \boldsymbol{V}_\mathrm{B} \end{bmatrix}. \qquad (17)$$

Here, the matrices $\boldsymbol{V}_X \in \mathbb{R}^{d\times c}$, $\boldsymbol{V}_X\boldsymbol{V}_\mathrm{F} \in \mathbb{R}^{d\times c}$, and $\boldsymbol{V}_X\boldsymbol{V}_\mathrm{B} \in \mathbb{R}^{d\times c}$ are used to compress the three feature groups.

An important difference from PCAPass is that $\boldsymbol{V}_X$ is computed separately via PCA from $\boldsymbol{V}_\mathrm{F}$ and $\boldsymbol{V}_\mathrm{B}$, meaning that $\boldsymbol{V}_X$ forms an orthogonal basis by itself, i.e., $\boldsymbol{V}_X^\mathsf{T}\boldsymbol{V}_X = \boldsymbol{I}_c$. Consequently, the column norms of $\boldsymbol{V}_X\boldsymbol{V}_\mathrm{F}$ are equal to the column norms in $\boldsymbol{V}_\mathrm{F}$, and similarly for $\boldsymbol{V}_X\boldsymbol{V}_\mathrm{B}$ and $\boldsymbol{V}_\mathrm{B}$. This can be demonstrated by considering the norm of the $i$th column in $\boldsymbol{V}_X\boldsymbol{V}_\mathrm{F}$:

$$\|\boldsymbol{V}_X\boldsymbol{V}_{\mathrm{F}i,:}\|_2^2 = \boldsymbol{V}_{\mathrm{F}i,:}^\mathsf{T}\boldsymbol{V}_X^\mathsf{T}\boldsymbol{V}_X\boldsymbol{V}_{\mathrm{F}i,:} = \boldsymbol{V}_{\mathrm{F}i,:}^\mathsf{T}\boldsymbol{I}_c\boldsymbol{V}_{\mathrm{F}i,:} = \|\boldsymbol{V}_{\mathrm{F}i,:}\|_2^2. \qquad (18)$$

Therefore, the ACC proportion vectors are

$$\mathbf{v}_X = \sum_{i=1}^{d} \left(\boldsymbol{V}_{Xi,:}\right)^2 = \mathbf{1}_c, \qquad \mathbf{v}_\mathrm{F} = \sum_{i=1}^{c} \left(\boldsymbol{V}_{\mathrm{F}i,:}\right)^2, \qquad \mathbf{v}_\mathrm{B} = \sum_{i=1}^{c} \left(\boldsymbol{V}_{\mathrm{B}i,:}\right)^2, \qquad (19)$$

where $\mathbf{v}_\mathrm{F} + \mathbf{v}_\mathrm{B} = \mathbf{1}_c$.

In Figure 14, we visualize the proportion vectors as heat maps. The heat maps on the left show the PCAPass vectors, $\mathbf{w}_X$, $\mathbf{w}_\mathrm{F}$, and $\mathbf{w}_\mathrm{B}$, while the heat maps on the right display the ACC vectors, $\mathbf{v}_\mathrm{F}$ and $\mathbf{v}_\mathrm{B}$. The colours in the heat maps represent the proportion of information derived from each feature group. The bright colours in the first two rows for PCAPass indicate that most of the information is captured from the features in $\boldsymbol{X}$ and $\boldsymbol{A}_\mathrm{F}\boldsymbol{X}$. In contrast, ACC captures more information from $\boldsymbol{A}_\mathrm{B}\boldsymbol{X}$, as evidenced by the more uniform colour distribution in its heat map.

We can further quantify this difference by computing the effective number of features extracted from each feature group. We denote this as $p_*^{(\mathrm{eff})}$, where the star is either $X$, F, or B for each respective feature group. These quantities are computed as the sum of each proportion vector:

$$p_X^{(\mathrm{eff})} = \sum_{k=1}^{p} w_{Xk}, \qquad p_\mathrm{F}^{(\mathrm{eff})} = \sum_{k=1}^{p} w_{\mathrm{F}k}, \qquad p_\mathrm{B}^{(\mathrm{eff})} = \sum_{k=1}^{p} w_{\mathrm{B}k}, \qquad (20)$$

for PCAPass, and

$$p_X^{(\mathrm{eff})} = \sum_{k=1}^{c} v_X = 256, \qquad p_\mathrm{F}^{(\mathrm{eff})} = \sum_{k=1}^{c} v_{\mathrm{F}k}, \qquad p_\mathrm{B}^{(\mathrm{eff})} = \sum_{k=1}^{c} v_{\mathrm{B}k}, \qquad (21)$$

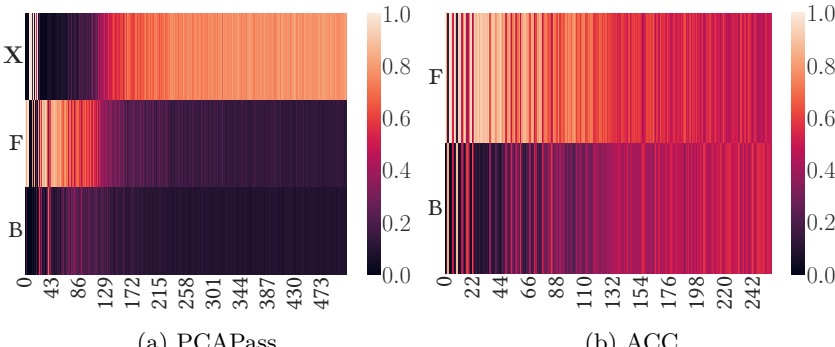

(a) PCAPass — (b) ACC

| Model | $p_X^{(\text{eff})}$ | $p_F^{(\text{eff})}$ | $p_B^{(\text{eff})}$ |
|---|---|---|---|
| PCAPass | 309 | 140 | 63 |
| ACC | 256 | 160 | 96 |

Figure 14: Visualization of the projection matrices used in the first message-passing iteration for PCAPass and ACC. Each column represents an embedding dimension, while each row corresponds to one of three feature groups: the input features $X$, the forward aggregation features $A_F X$, and the backward aggregation features $A_B X$. The colour indicates the proportion of each feature group represented in each embedding dimension. These proportions are calculated using Equations 16 and 19.

Table 4: The effective number of embedding dimensions used per feature group after one message-passing iteration using ACC and PCAPass. The effective dimensions are calculated using Equations 20 and 21.

for ACC. Note that $p_X^{(\text{eff})} = c = 256$ since ACC always includes $c$ features per message-passing iteration.

The effective number of embedding dimensions is shown in Table 4. Compared to PCAPass, ACC effectively uses 53 fewer features from $X$ and 33 more features from $A_B X$, which represents more than a 50% increase compared to 63 for PCAPass. The inclusion of these class-informative features explains the higher accuracy achieved by message aggregation compared to embedding aggregation on the Squirrel dataset.

## G   Experimental investigation of rank deficiency in self-supervised graph neural networks

### G.1   Singular value spectra and graph alignment experiments

In this section, we present experimental results supporting the discussion in Section 6 on rank deficiency in self-supervised graph neural networks (SSGNNs). As described there, we evaluate SSGNN versions of PCAPass and ACC, trained by minimizing the GraphMAEv2 loss (Hou et al., 2023), and assess their behaviour after 0, 20, and 200 training epochs.

We first revisit the experiments from Section B and compute the singular value spectra of the resulting embeddings for $K = 4$ and $K = 10$. The results are shown in Figure 15. Comparing the spectra of PCAPass-GNN to those of the linear PCAPass model in Figure 7, we observe that PCAPass-GNN does not exhibit rank deficiency, as there is no clear spectral gap. This confirms that the inclusion of non-linearities in the model prevents the explicit rank collapse seen in the linear setting.

The next question is whether the absence of rank deficiency in PCAPass-GNN implies improved embedding quality. To investigate this, we repeat the graph alignment experiments from Section 5.1 using the learned embeddings of both PCAPass-GNN and ACC-GNN.

The results are shown in Figure 16. Notably, we observe a significant increase in variance across random seeds for both PCAPass-GNN and ACC-GNN, compared to their linear counterparts. This increased variability stems from the random initialization of the learnable weight matrices introduced in Equations 12 and 13, which introduces greater sensitivity to seed selection.

In terms of accuracy, PCAPass-GNN substantially outperforms the original linear PCAPass model (cf. Figure 5), suggesting that resolving rank deficiency does indeed improve embedding quality. However, performance degrades with increased training: PCAPass-GNN trained for 200 epochs performs poorly across

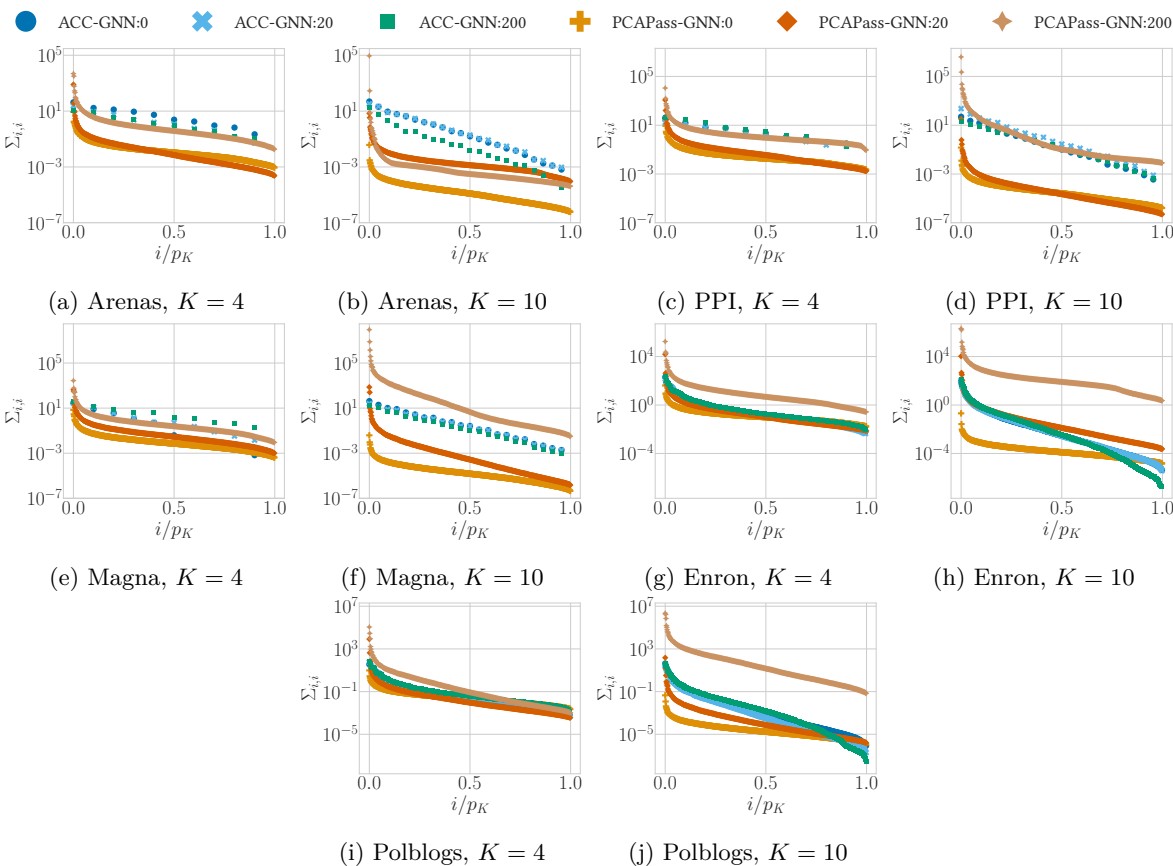

Figure 15: The singular values spectrum for the SSGNN versions of ACC and PCAPass on all graph alignment datasets using $K = 4$ and $K = 10$. The colours correspond to 0, 20 and 200 training epochs for ACC-GNN and PCAPass-GNN respectively. The y-axes show the singular values, and the x-axes their index in descending order, normalized using the number of embedding dimensions $p_K$.

all datasets. Additionally, as the number of message-passing iterations $K$ increases, PCAPass-GNN initially improves, reaching a peak accuracy, but then consistently declines beyond a certain point.

In contrast, ACC-GNN exhibits greater robustness. Its accuracy does not degrade with increasing training epochs or message-passing iterations $K$ on the undirected datasets. The only dataset where increased training leads to a noticeable drop in accuracy is Enron, see ACC-GNN:200. On Polblogs, accuracy decreases with increasing $K$, reflecting a trend also observed for the linear ACC model, as discussed in Section D.1.

Despite the absence of a clear spectral gap, we can still relate the behaviour of the graph alignment accuracies to the singular value spectra of PCAPass-GNN and ACC-GNN. To this end, we consider the *condition number* $\kappa(\boldsymbol{Z})$ of the embedding matrix $\boldsymbol{Z}$, defined as the ratio between its largest and smallest singular values:

$$\kappa(\boldsymbol{Z}) = \frac{\sigma_{\max}(\boldsymbol{Z})}{\sigma_{\min}(\boldsymbol{Z})}. \tag{22}$$

The condition number is closely related to the numerical sensitivity of matrix operations, particularly matrix inversion (Higham & Al-Mohy, 2010, Ch. 5.8). In general, higher condition numbers indicate greater sensitivity and potential numerical instability. While rank deficiency often leads to a high condition number, the converse does not necessarily hold: a matrix can have a large condition number even in the absence of an explicit spectral gap.

Table 5 reports the condition numbers of PCAPass-GNN and ACC-GNN for $K = 4$ and $K = 10$, across training epochs 0, 20, and 200, and for each dataset. For comparison, we also include the condition numbers of the linear PCAPass and ACC embeddings.

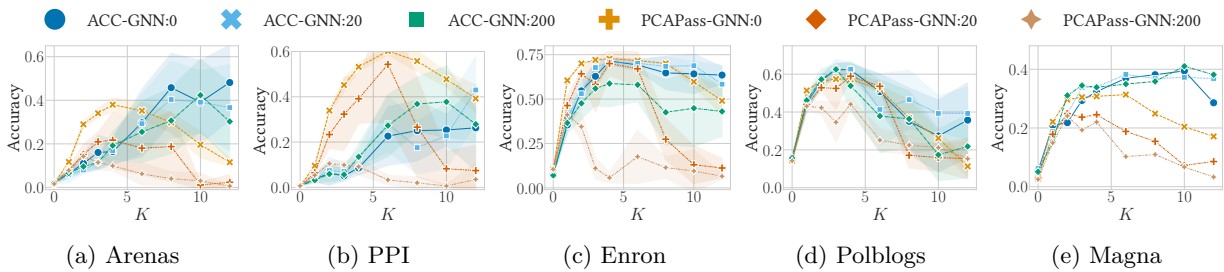

Figure 16: SSGNN versions of ACC and PCAPass for graph alignment with 15% noise edges. The x-axis shows the number of message-passing iterations $K$. The colour shades represent distinct number of training epochs. Markers and shaded areas represent the average and standard deviation over 3 seeds.

The general trend is that, for a fixed value of $K$, embeddings with lower condition numbers $\kappa$ tend to yield higher graph alignment accuracies. This pattern is clearly observed for ACC and PCAPass on the undirected datasets (Arenas, PPI, and Magna), where ACC exhibits significantly lower $\kappa$ values alongside substantially higher alignment performance. A similar relationship holds for Enron and Polblogs at $K = 4$. At $K = 10$, however, the condition numbers and alignment accuracies of ACC and PCAPass become more comparable, reflecting the effects of deeper message-passing and greater compression. These observations align with the rank deficiency analysis presented in the main paper.

A similar trend appears in the GNN variants. For 200 training epochs, PCAPass-GNN typically exhibits much higher condition numbers than ACC-GNN, and correspondingly lower alignment accuracies. At 0 epochs of training, their $\kappa$ values are more similar, and so are their accuracies, suggesting that weight optimization in PCAPass-GNN amplifies instability in the embeddings.

Nonetheless, there are notable exceptions to this trend. For instance, on Polblogs at $K = 10$, PCAPass-GNN with 0 epochs has a much lower condition number than ACC-GNN, but does not outperform it in accuracy. Additionally, both ACC and ACC-GNN tend to have higher condition numbers at $K = 10$ than at $K = 4$, yet their accuracies also increase, highlighting that $\kappa$ is not a sufficient standalone predictor of embedding quality.

In conclusion, while the SSGNN variant of PCAPass is not explicitly rank-deficient, there appears to be a correlation between information redundancy induced by its embedding aggregation and concatenation strategy, elevated condition numbers, and reduced embedding quality. These issues are mitigated by ACC's message aggregation approach, both in linear and non-linear settings. However, analysing the non-linear GNN case remains significantly more complex and calls for deeper theoretical and empirical investigation in future work.

### G.2    Node classification results for ACC-GNN and PCAPass-GNN

We also evaluate node classification accuracy for both ACC-GNN and PCAPass-GNN. Results using a gradient boosting classifier are reported in Table 6, and those using logistic regression are shown in Table 7.

Table 5: The condition number of the embedding matrices for ACC and PCAPass, and the ACC-GNN and PCAPass-GNN models for 0, 20 and 200 epochs of training. Displayed values are $\log_{10} \kappa(\boldsymbol{Z})$.

(a) $K = 4$

| MODEL | ARENAS | PPI | MAGNA | ENRON | POLBLOGS |
|---|---|---|---|---|---|
| ACC | 1.83 | 2.23 | 1.84 | 2.98 | 3.41 |
| PCAPASS | 8.61 | 8.82 | 8.83 | 9.16 | 9.54 |
| ACC-GNN:0 | 2.29 | 2.09 | 4.62 | 4.49 | 4.72 |
| PCAPASS-GNN:0 | 3.99 | 4.06 | 4.23 | 3.38 | 3.63 |
| ACC-GNN:20 | 2.67 | 1.96 | 3.27 | 4.50 | 5.12 |
| PCAPASS-GNN:20 | 6.53 | 5.82 | 5.64 | 6.32 | 7.36 |
| ACC-GNN:200 | 1.98 | 2.38 | 2.26 | 4.35 | 4.63 |
| PCAPASS-GNN:200 | 5.41 | 5.09 | 5.48 | 5.81 | 8.03 |

(b) $K = 10$

| MODEL | ARENAS | PPI | MAGNA | ENRON | POLBLOGS |
|---|---|---|---|---|---|
| ACC | 4.34 | 4.70 | 4.26 | 4.41 | 5.20 |
| PCAPASS | 9.35 | 9.32 | 9.74 | 3.06 | 3.85 |
| ACC-GNN:0 | 4.87 | 5.14 | 4.34 | 7.33 | 7.71 |
| PCAPASS-GNN:0 | 4.80 | 4.96 | 4.94 | 4.14 | 4.60 |
| ACC-GNN:20 | 4.54 | 5.46 | 4.14 | 7.46 | 8.14 |
| PCAPASS-GNN:20 | 4.88 | 7.81 | 8.70 | 7.65 | 7.96 |
| ACC-GNN:200 | 5.78 | 4.71 | 4.24 | 8.88 | 9.16 |
| PCAPASS-GNN:200 | 9.34 | 8.69 | 9.46 | 5.95 | 7.52 |

Table 6: Gradient boosting node classification results for SSGNN versions of ACC and PCAPass. The results for ACC and the best performing SSGNNs from Table 1 are included for reference. The highest accuracy is highlighted in bold.

| Model | Chameleon Accuracy | Time | Squirrel Accuracy | Time | Roman Empire Accuracy | Time | Arxiv Year Accuracy | Time | Snap-Patents Accuracy | Time |
|---|---|---|---|---|---|---|---|---|---|---|
| No model, $\mathbf{X}$ | $64.6 \pm 2.3$ | 0ms | $52.1 \pm 1.4$ | 0ms | $70.4 \pm 0.7$ | 0ms | $44.7 \pm 0.2$ | 0ms | $56.1 \pm 0.1$ | 0ms |
| CCA-SSG[1] | $71.0 \pm 2.1$ | 4s | $59.9 \pm 1.2$ | 7s | $63.7 \pm 0.7$ | 10s | $48.4 \pm 0.2$ | 1m 11s | $56.1 \pm 0.0$ | 1h 40m |
| GraphMAEv2-GS[2] | $74.1 \pm 2.3$ | 29s | $57.0 \pm 1.4$ | 1m 6s | $80.0 \pm 0.7$ | 1m 35s | $46.3 \pm 0.3$ | 10m 46s | $53.8 \pm 0.1$ | 18h |
| ACC | $\mathbf{76.6 \pm 1.9}$ | 1s | $\mathbf{71.5 \pm 1.3}$ | 1s | $\mathbf{81.5 \pm 0.6}$ | 432ms | $\mathbf{49.4 \pm 0.3}$ | 1s | $\mathbf{62.6 \pm 0.1}$ | 27s |
| ACC-GNN | $73.8 \pm 1.9$ | 906ms | $66.7 \pm 1.4$ | 1s | $78.0 \pm 0.8$ | 1s | $47.9 \pm 0.2$ | 8s | $62.3 \pm 0.1$ | 11m 23s |
| PCAPass-GNN | $66.1 \pm 2.1$ | 989ms | $45.0 \pm 1.5$ | 1s | $66.2 \pm 0.9$ | 2s | $45.2 \pm 0.3$ | 16s | $60.0 \pm 0.1$ | 16m 17s |

[1] Zhang et al. (2021)   [2] Hou et al. (2023)

Table 7: Logistic regression node classification results for SSGNN versions of ACC and PCAPass. The results for ACC and the best performing SSGNNs from Table 1 are included for reference. The highest accuracy is highlighted in bold.

| Model | Chameleon Accuracy | Time | Squirrel Accuracy | Time | Roman Empire Accuracy | Time | Arxiv Year Accuracy | Time | Snap-Patents Accuracy | Time |
|---|---|---|---|---|---|---|---|---|---|---|
| No model, $\mathbf{X}$ | $52.3 \pm 2.2$ | 0ms | $35.3 \pm 1.4$ | 0ms | $69.8 \pm 0.7$ | 0ms | $43.4 \pm 0.2$ | 0ms | $50.7 \pm 0.1$ | 0ms |
| CCA-SSG[1] | $59.5 \pm 2.7$ | 4s | $41.6 \pm 1.4$ | 7s | $70.0 \pm 0.7$ | 10s | $\mathbf{50.9 \pm 0.3}$ | 1m 11s | $55.3 \pm 0.1$ | 1h 40m |
| GraphMAEv2-GS[2] | $\mathbf{70.6 \pm 2.3}$ | 29s | $\mathbf{48.6 \pm 1.6}$ | 1m 6s | $\mathbf{80.2 \pm 0.7}$ | 1m 35s | $46.9 \pm 0.3$ | 10m 46s | $54.7 \pm 0.0$ | 18h |
| ACC | $60.7 \pm 2.5$ | 1s | $44.1 \pm 1.4$ | 1s | $79.3 \pm 0.6$ | 432ms | $49.4 \pm 0.3$ | 1s | $56.6 \pm 0.0$ | 27s |
| ACC-GNN | $57.8 \pm 3.0$ | 906ms | $42.2 \pm 1.7$ | 1s | $77.7 \pm 0.6$ | 1s | $48.7 \pm 0.4$ | 8s | $\mathbf{61.3 \pm 0.1}$ | 11m 23s |
| PCAPass-GNN | $56.0 \pm 2.9$ | 989ms | $39.0 \pm 2.0$ | 1s | $74.4 \pm 0.8$ | 2s | $47.7 \pm 0.4$ | 16s | $60.6 \pm 0.1$ | 16m 17s |

[1] Zhang et al. (2021)   [2] Hou et al. (2023)

For ease of comparison, we also include the best-performing results from the main paper. All experiments were conducted with 20 training epochs and consistent hyperparameter settings: $K = 2$ and $p_{max} = 512$.

Consistent with earlier findings, ACC-GNN generally outperforms PCAPass-GNN across datasets, suggesting that message aggregation produces higher-quality embeddings than embedding aggregation, even in the presence of non-linearity. Furthermore, although self-supervised graph neural networks (SSGNNs) are theoretically more expressive, they do not outperform ACC in practice under default hyperparameters. Due to their reliance on iterative gradient descent, these models are also significantly less efficient, particularly on large-scale datasets such as Snap-Patents.

## H Hyperparameter tuning of self-supervised graph neural network baselines

For the node classification benchmarks presented in Section 5.2, we fix the number of message-passing iterations to $K = 2$ and the embedding dimension to $p = 512$ across all embedding models, while keeping all other hyperparameters at their default settings. This ensures a fair comparison by standardizing two of the most influential factors across models.

Since $K$ and $p$ have a well-documented impact on embedding quality, as demonstrated for ACC in Section D, using consistent values for these parameters eliminates variation due to their influence. We selected $K = 2$ and $p = 512$ based on their frequent use in prior work (Veličković et al., 2019; Zhang et al., 2021; Thakoor et al., 2022; Hou et al., 2022; Wang et al., 2023). For all other hyperparameters, which vary between models, we use the authors' default configurations, as these have typically been optimized and validated during the original model development.

An alternative approach would be to perform hyperparameter tuning and select the best configuration for each model and dataset. However, there are several reasons to avoid this strategy. First, hyperparameter tuning is notoriously difficult in real-world unsupervised learning scenarios (Ma et al., 2023). This difficulty

arises from the fact that tuning typically relies on ground-truth labels to guide the selection process. In unsupervised learning, however, such label information is not available by design. Indeed, the absence of labels is precisely why unsupervised methods are used in the first place. Consequently, evaluating unsupervised embedding models under their default hyperparameters provides a more realistic and practical assessment of their utility in real-world applications.

Second, hyperparameter tuning is computationally expensive. Each self-supervised graph neural network (SSGNN) includes numerous hyperparameters, and evaluating different combinations requires training and validating the model multiple times. Ideally, this process must also be repeated over several random seeds to account for initialization variance. As shown in Table 3, training SSGNNs is significantly more time-consuming than training ACC. Thus, comprehensive hyperparameter tuning would require substantial compute resources and cloud GPU hours, which are unfortunately beyond the financial means of the authors.

While comprehensive hyperparameter tuning is infeasible, we conduct a limited tuning experiment to estimate the potential accuracy gains that SSGNNs might achieve with optimized settings. Specifically, we tune the embedding dimension $p$, which we previously found to significantly influence node classification accuracy for ACC (Figure 10), and the number of training epochs, a common tuning parameter in SSGNN studies (Veličković et al., 2019; Hassani & Khasahmadi, 2020; Zhang et al., 2021; Hou et al., 2022; 2023; Wang et al., 2023). To reduce computational overhead, we use logistic regression rather than gradient boosting for evaluation, as the latter is substantially more expensive to train and validate.

The classification accuracy results for each baseline model and dataset are presented in Figures 17 and 18. Each coloured line represents a different embedding dimension $p$, and the x-axis shows the number of training epochs. The orange marker highlights the best-performing configuration observed during the sweep, while the black dashed line indicates the accuracy reported in Table 3 using $p = 512$ and the model's default number of training epochs (see Section C.3). Figures 19 and 20 show the corresponding loss values across training.

The primary observation from the tuning experiment is that, for the majority of models and datasets, accuracy tends to increase with the embedding dimension $p$. This is expected: a larger embedding space allows the classifier to model more complex decision boundaries, reducing bias. The accompanying increase in variance is mitigated by the large number of training samples and the use of regularization in the classifier (Bishop, 2006, Ch. 3.2). However, increasing $p$ also leads to greater computational cost. These findings support our decision to evaluate all models using a common embedding dimensionality of $p = 512$, as comparing models with different $p$ values would introduce unfair advantages to those using higher-dimensional embeddings.

That said, we also observe that for GAE, GraphMAE, and GraphMAEv2, performance degrades when $p = 2048$ on the Chameleon and Squirrel datasets. In these cases, accuracy drops noticeably compared to $p = 512$, and this drop is accompanied by a sharp increase in loss values. We speculate that this behaviour stems from these models' reliance on autoencoding losses, which may overfit or become unstable when the embedding space is too large relative to the dataset size. This hypothesis is supported by the fact that Chameleon and Squirrel are the two smallest datasets in terms of node count. However, further investigation is required to confirm the underlying cause.

Regarding the number of training epochs, we find no consistent benefit from deviating from the default values. For most models with $p = 512$, accuracy either peaks near the default epoch setting or plateaus around the same value. The exceptions are BGRL and MVGRL, where accuracy improves with longer training. Notably, their default epoch values—10,000 for BGRL and 3,000 for MVGRL—exceed the maximum number of epochs tested in our sweep, suggesting that the defaults are already well-chosen in these cases.

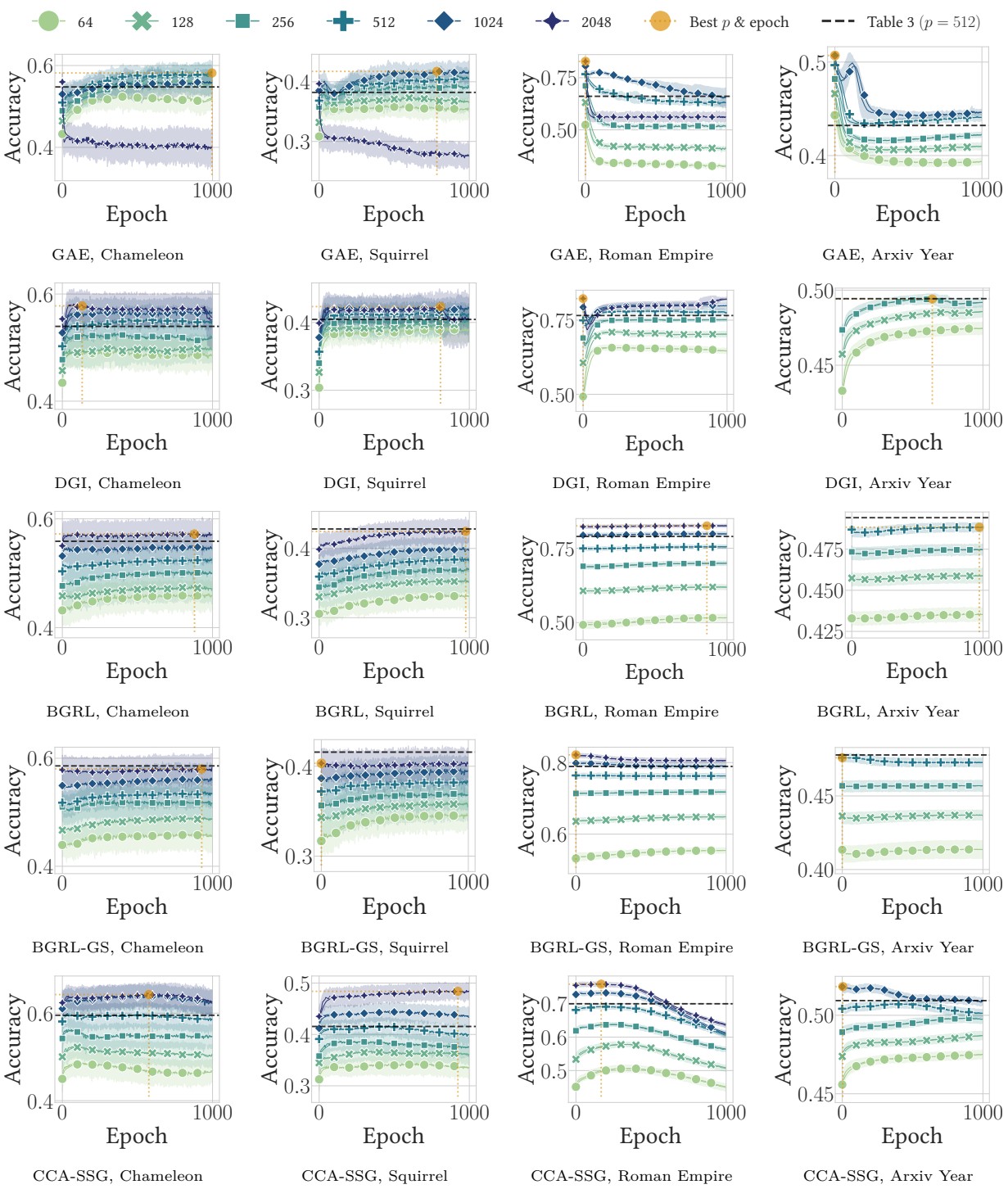

Figure 17: Node classification accuracy across training epochs for GAE, DGI, BGRL, BGRL-GS, and CCA-SSG. Each subplot corresponds to a dataset, while line colours and marker shapes represent the embedding dimension $p$. The orange marker indicates the highest observed accuracy across all settings. The black dashed line shows the accuracy reported in Table 3 for $p = 512$ and the model's default number of training epochs. Shaded areas represent standard deviations computed over five repeated 5-fold cross-validation runs for the classifier, using five different random seeds for the embedding model.

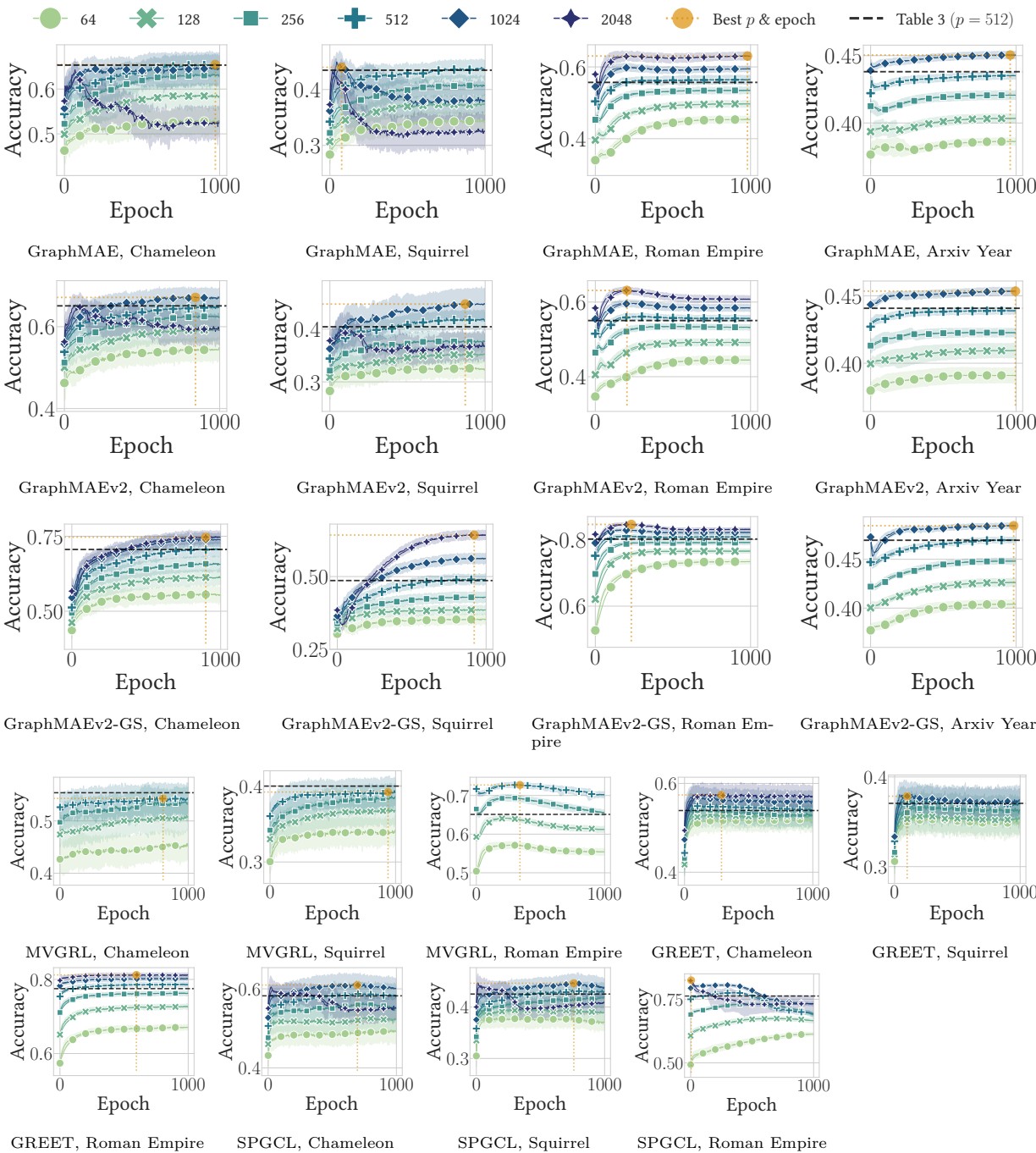

Figure 18: Node classification accuracy across training epochs for GraphMAE, GraphMAEv2, GraphMAEv2-GS, MVGRL, GREET, and SPGCL. Each subplot corresponds to a dataset, while line colours and marker shapes represent the embedding dimension $p$. The orange marker indicates the highest observed accuracy across all settings. The black dashed line shows the accuracy reported in Table 3 for $p = 512$ and the model's default number of training epochs. Shaded areas represent standard deviations computed over five repeated 5-fold cross-validation runs for the classifier, using five different random seeds for the embedding model.

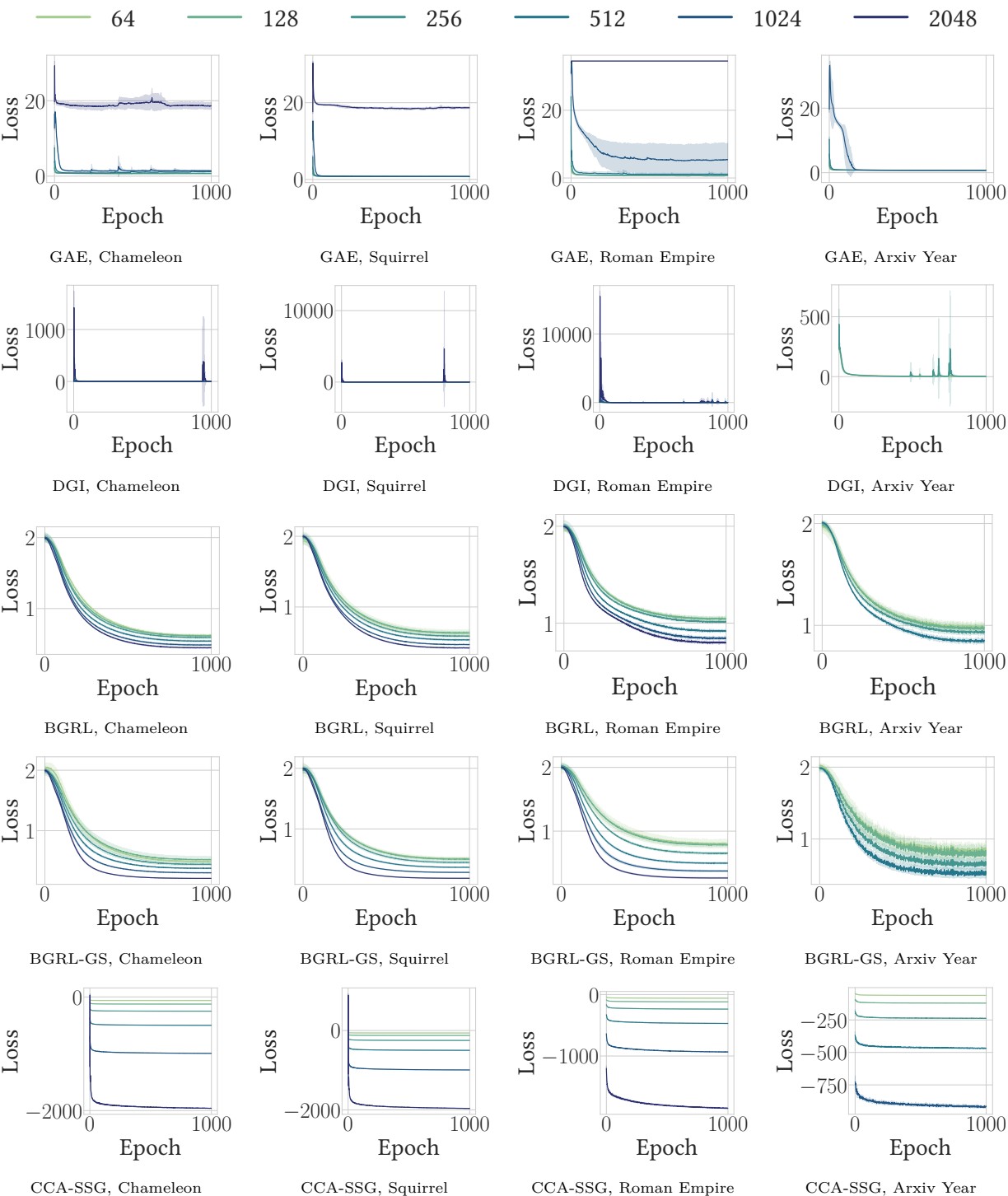

Figure 19: Loss curves across training epochs for GAE, DGI, BGRL, BGRL-GS, CCA-SSG. Each subplot corresponds to a dataset, while line colours represent the embedding dimension $p$. Shaded areas represent standard deviations over 5 different seeds.

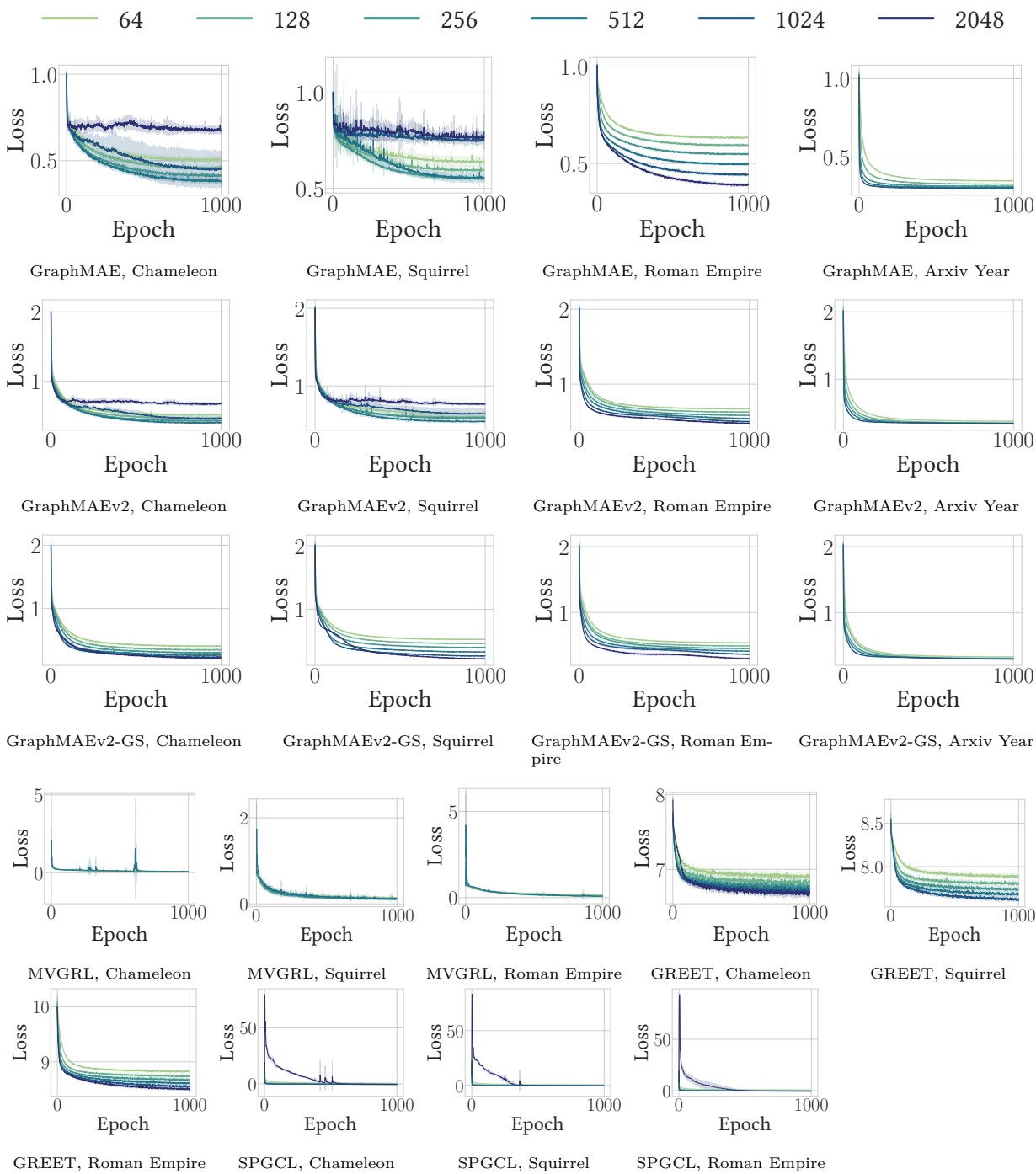

Figure 20: Loss curves across training epochs for GraphMAE, GraphMAEv2, GraphMAEv2-GS, MVGRL, GREET, and SPGCL. Each subplot corresponds to a dataset, while line colours represent the embedding dimension $p$. Shaded areas represent standard deviations over 5 different seeds.

