# OpenReview forum: "Full-Rank Unsupervised Node Embeddings for Directed Graphs via Message Aggregation"
_TMLR — Accepted by TMLR_

### Review · Reviewer_7mXH · 2024-09-23

**Summary Of Contributions:**

In this paper, the authors aim to solve the rank deficiency in one previous method PCAPass. It first proves the rank deficiency existed in PCAPass and then improves it to eliminate this effect. Experiments show the effectiveness of the proposed method compared to PCAPass.

**Audience:**

Yes

**Claims And Evidence:**

No

**Requested Changes:**

Please kindly refer to the Strengths And Weaknesses section. I suggest the authors tune self-supervised graph neural networks.

**Strengths And Weaknesses:**

**Strengths**
- This paper is well-written and the presentation is clear.
- The proposed method solved the rank deficiency issue in PCAPass and can learn more stable node representations.

**Weakness**
- Traditional self-supervised graph neural networks already perform very well. This work is an improvement on PCAPass [1]. So from my perspective, even the groundwork for this paper, PCAPass, has *little practical application*.
- The experiment settings in 'Section 5.2 Node classification: ACC vs self-supervised graph neural networks' do not make sense to me. The authors claim that 'We do not perform hyperparameter tuning, as it is well-established that SSGNNs can surpass linear models
like ACC in accuracy with sufficient tuning.............' However it is not practical for them to use the default hyperparameter set on all datasets. The comparison is not fair.

[1] Sadowski, Krzysztof, Michał Szarmach, and Eddie Mattia. "Dimensionality reduction meets message passing for graph node embeddings." arXiv preprint arXiv:2202.00408 (2022).

---

> ### Author Response · Authors · 2025-02-20
> **Response to Reviewer 7mXH**
>
> Thank you for taking the time to review our paper and for recognizing our work on addressing the issue of rank deficiency in unsupervised learning of node embeddings. We appreciate your positive remarks about the clarity and presentation of our method. We now address the concerns raised in your review with individual comments.

---

> > ### Author Response · Authors · 2025-02-20
> > **Practical Applicability**
> >
> > We respectfully disagree with the assertion that our work has limited practical application, and offer three points to support our view:
> >
> > 1. **Scalability**: As highlighted in the final sentence of our introduction, ACC’s primary advantage over self-supervised graph neural networks (SSGNNs) is its scalability. In Section 5.2 (p. 12), we provide a detailed analysis of ACC's efficiency, noting:
> >
> >    > "On the Arxiv Year dataset, with 1 million edges, ACC is over 70 times faster than CCA-SSG, the fastest SSGNN. Moreover, whereas most SSGNNs take a full day or more to run on Snap Patents, the largest dataset with 14 million edges, ACC requires only 27 seconds."
> >
> >    These substantial speedups underscore ACC’s practical relevance, particularly for large-scale graphs where computational efficiency is paramount.
> >
> > 2. **Applicability beyond PCAPass**: While our work addresses the rank deficiency in a specific method (PCAPass), its relevance extends to a broader class of node embedding approaches. As discussed in Section 2 and revisited in Section 3.2 of [MUSAE], concatenation is a well-established technique for generating multi-scale node embeddings (e.g., [LINE], [GraRep], [Walklets], and [GraphWave]). Thus, although PCAPass is one recent example (2022), our work is relevant for a wide range of methods that produce multi-scale embeddings via concatenation, and we have updated the second paragraph on page 2 of our paper to reflect this broader relevance.
> >
> > 3. **Directed Graphs**: Our work fills an important gap in the relatively underexplored area of node embeddings for directed graphs. ACC contributes a scalable, rank-sufficient solution tailored to the unique challenges posed by these graphs, expanding the toolbox available to practitioners working in this space.
> >
> > References
> >
> >     [MUSAE] Benedek Rozemberczki et al., Multi-Scale Attributed Node Embedding, Journal of Complex Networks, Volume 9, Issue 2, April 2021, https://doi.org/10.1093/comnet/cnab014
> >     [LINE] Jian Tang et al. Line: Large-Scale Information Network Embedding. In WWW ’15, 2015. URL https://doi.org/10.1145/2736277.2741093.
> >     [GraRep] Shaosheng Cao et al. Grarep: Learning graph representations with global structural information. In CIKM ’15, pp. 891–900, 2015. https://doi.org/10.1145/2806416.2806512.
> >     [Walklets] Bryan Perozzi et al. Don’t walk, skip! online learning of multi-scale network embeddings. In ASONAM ’17, pp. 258–265, 2017. https://doi.org/10.1145/3110025.3110086.
> >     [GraphWave] Claire Donnat et al. Learning Structural Node Embeddings Via Diffusion Wavelets. In KDD’18, pp. 1320–1329, 2018. https://doi.org/10.1145/3219819.3220025.

---

> > ### Author Response · Authors · 2025-02-20
> > **Fairness of Comparison to SSGNNs**
> >
> > We would like to clarify our position regarding the comparison between ACC and SSGNNs. While we understand the reviewer's concern about hyperparameter tuning for the baselines, we maintain that our approach is fair and appropriate for this paper's scope. We divide our reponse into two points.
> >
> >
> > ### 1. The Challenge of hyperparameter tuning in unsupervised embedding learning
> >
> > Our work focuses on unsupervised embedding learning, where node labels are not assumed to be available during training. In a fully supervised setting, one would naturally tune hyperparameters using validation labels. However, in real-world scenarios where such labels are unavailable, identifying optimal hyperparameters becomes highly challenging.
> >
> > As we mentioned on page 11 (in the original paper before revision):
> >
> > > "... tuning hyperparameters in unsupervised settings is known to be difficult (Ma et al., 2023) due to the absence of validation data with ground-truth labels. [...] Without access to ground-truth information, identifying the optimal hyperparameter configuration becomes highly challenging."
> >
> >
> > For this reason, we do not tune hyperparameters for any model in our evaluation. Instead, we use default hyperparameters provided by the original authors. These defaults are presumably chosen to yield general-purpose embeddings suitable for a range of downstream tasks. We argue that this approach better reflects practical usage than attempting to fine-tune models on a per-dataset basis, which is infeasible in a true unsupervised setting.
> >
> > ### 2. Consistency in Experimental Setup
> >
> > We ensure fairness by keeping key hyperparameters *consistent* across all models. Specifically, we use $K=2$ message-passing iterations and set the embedding dimensionality to $p=512$, values commonly used in the literature. These settings are not necessarily optimal for ACC, as demonstrated in Figure 10 of the Appendix, further reinforcing that we do not tune ACC for maximum performance.
> >
> > Beyond $K$ and $p$, SSGNNs require additional hyperparameters related to loss functions and optimizers. Tuning these per dataset is both computationally expensive and incompatible with our unsupervised setting. In contrast, ACC does not rely on such additional hyperparameters, making direct tuning-based comparisons impractical.
> >
> > A truly fair evaluation would consider the computational cost of tuning (in GPU hours and human effort) across models. However, this evaluation is beyond the scope of our paper, as our primary focus is addressing the rank deficiency problem.
> >
> > Instead, we believe that using consistent values for $K$ and $p$ across models, alongside default optimizer and loss function hyperparameters, results in a fair and practical evaluation. This approach is particularly justifiable given the common practice in the literature of using default hyperparameters for baselines while tuning one's own model with ground-truth labels, as seen in works such as [GraphMAEv2], [BGRL], and [SPGCL].
> >
> >
> > ### Summary
> >
> > In summary, we believe that our experimental setup is fair and representative of real-world scenarios where hyperparameter tuning is infeasible. To further improve clarity, we have updated the introduction section of our paper to explicitly state that we do not perform hyperparameter tuning.
> >
> > We sincerely appreciate the reviewer’s feedback, which has helped us refine our presentation.
> >
> >
> > References
> >
> >     [BGRL] Shantanu Thakoor et al. Large-Scale Representation Learning on Graphs via Bootstrapping. ICLR 2022. URL: https://doi.org/10.48550/arXiv.2102.06514
> >     [GraphMAEv2] Zhenyu Hou et al. GraphMAE2: A Decoding-Enhanced Masked Self-Supervised Graph Learner. WWW 2023, pp. 737–746. URL: https://doi.org/10.1145/3543507.3583379
> >     [SPGCL] Haonan Wang et al. Single-Pass Contrastive Learning Can Work for Both Homophilic and Heterophilic Graphs. TMLR, 2023. URL: https://openreview.net/forum?id=244KePn09i

---

### Review · Reviewer_iHLV · 2024-10-27

**Summary Of Contributions:**

The paper:

Identifies and analyzes a fundamental flaw (rank deficiency) in linear message-passing models that use embedding aggregation and concatenation, particularly in PCAPass.

Introduces ACC (Aggregate, Compress, Concatenate).

Provides theoretical analysis of why rank deficiency occurs in previous approaches and how ACC resolves this issue.

Demonstrates empirically that ACC is 70-270x faster than existing methods while maintaining competitive or superior accuracy on large-scale graphs.

**Audience:**

Yes

**Broader Impact Concerns:**

N/A.

**Claims And Evidence:**

No

**Requested Changes:**

Consider including guidelines for selecting optimal hyperparameters.

Include more detailed analysis of how different compression strategies might affect performance.

Verify the proof of rank deficiency in general case.

**Strengths And Weaknesses:**

Weakness:

The paper proves redundancy through a simple example of duplicated columns, but doesn't rigorously prove why PCAPass necessarily leads to rank deficiency in the general case. It leaves open the possibility that PCAPass's rank deficiency might be implementation-dependent rather than fundamental.

The hyperparameters chosed in the paper could be further discussed. For example, do optimal hyperparameter values differ between tasks (e.g., graph alignment vs. node classification)?

---

> ### Author Response · Authors · 2025-02-20
> **Response to Reviewer iHLV**
>
> Thank you for taking the time to review our paper and for recognizing our work on addressing the issue of rank deficiency in unsupervised learning of node embeddings. We appreciate your positive remarks about the clarity and presentation of our method. We now address the concerns raised in your review.
>
>
> ### Rank Deficiency of PCAPass in the General Case
>
> As correctly pointed out, we used simplifying assumptions in our demonstration of the rank deficiency of the PCAPass embeddings in Section 3.2. Specifically, we initially ignored the effect of the compression matrix in the first part, highlighting instead the duplicated feature columns, and then discussed the impact of compression separately. This structure was intended to provide a clear and fluid presentation of the redundancy and rank issues.
>
> However, we recognize the reviewer’s point that this approach could leave open the possibility that the compression method could influence the rank deficiency. In response, we have updated Section 3.2 to demonstrate that rank deficiency is inherent to the PCAPass algorithm, regardless of the specific method used for dimensionality reduction. The revised section now provides a proof that shows PCAPass embeddings will be rank-deficient in the general case, irrespective of the compression technique.
>
> While this new presentation is more technical, we believe it strengthens our paper, and we thank the reviewer for this valuable feedback!
>
>
> ### Effect of Hyperparameters
>
> For a discussion on the effect of ACC’s hyperparameters, we would like to direct the reviewer to Section D of our appendix. In this section, we analyse and discuss the impact of both of ACC’s hyperparameters, $K$ and $p_{\text{max}}$, on graph alignment and node classification tasks.
>
> With respect to selecting optimal hyperparameters, we note that achieving optimal tuning can be challenging in the unsupervised learning setting. As a result, we use a consistent number of message-passing iterations and embedding dimensions across all models, specifically $K=2$ and $p=512$. As shown in Figure 10 of the Appendix, these settings are not necessarily optimal for ACC’s node classification accuracy.
>
> We have updated Section 5.2 to clarify our choice of hyperparameters for the benchmark, and our rationale for not performing extensive hyperparameter tuning.

---

### Review · Reviewer_DC2u · 2025-02-15

**Summary Of Contributions:**

This paper identifies the rank deficiency problem in existing linear models. This paper presents formal proof of how embedding aggregation leads to rank deficiency.

Then this paper introduces ACC (Aggregate, Compress, Concatenate), a novel linear message-passing model designed to overcome the issue of rank deficiency in node embeddings for directed graphs. The proposed ACC ensures embeddings remain full rank, improving alignment and classification tasks without additional computational cost.

**Audience:**

Yes

**Claims And Evidence:**

Yes

**Requested Changes:**

1. discuss the time complexity of the proposed method
2. discuss the inductive setting

**Strengths And Weaknesses:**

### Strengthnesses:
1. This paper provides novel theoretical discussion about the rank deficiency due to repeated embedding aggregation and concatenation in existing linear message-passing models (e.g., PCAPass). And it derives the lower bound $\rho_k \geq  (2^k − k − 1)d$
2. The proposed ACC novelly shifts the aggregation process from embedding aggregation to message aggregation, preventing redundancy and ensuring full-rank embeddings.
3. The empirical evaluation on graph alignment and node classification shows both the effectiveness and the computational efficiency of the propose ACC.

### Weaknesses:
1. This paper adopts the transductive setting, where the entire graph is available during training. However, many real-world applications involve new nodes appearing dynamically such as social networks and recommendation systems.
2. The time complexity analysis of the proposed method is missing.

---

> ### Author Response · Authors · 2025-02-20
> **Response to Reviewer DC2u**
>
> Thank you for taking the time to review our paper and for recognizing the novelty of our work. We also greatly appreciate your positive feedback regarding the effectiveness and efficiency of ACC.
>
> We agree that our paper can be strengthened by including a time complexity analysis of ACC and a discussion on its applicability to the inductive setting. To address these points, we have added Section 4.1 (page 10) to the revised paper, which covers both topics.
>
> In summary, the time complexity of ACC is determined by four variables: the number of nodes ($n$), the number of edges ($m$), and the dimensionalities of the input matrix ($d$) and the message matrices ($c$). The overall complexity of ACC is $O(nd^2 + Kmc + Knc^2)$. The first term, $O(nd^2)$, arises from the initial PCA compression of the input features. The second term, $O(mc)$, corresponds to the sparse-dense matrix multiplications performed for message aggregation, and the final term, $O(nc^2)$, accounts for the PCA compression of the message matrices, both of which are repeated for $K$ iterations.
>
> Additionally, ACC can be extended to the inductive setting using a GraphSAGE-inspired approach [GraphSAGE]. However, due to ACC’s high efficiency, inductive learning is primarily relevant for extremely large graphs. In many cases, retraining ACC on the full graph remains computationally feasible and may even be preferable. Given this, we leave a detailed empirical investigation of ACC in the inductive setting to future work.
>
>
> ##### Reference
>
>     [GraphSAGE] William L. Hamilton et al. Inductive Representation Learning on Large Graphs. In NeurIPS ’17, 2017. https://arxiv.org/abs/1706.02216.

---

### Author Response · Authors · 2025-02-20
**Authors' Response to Reviewer Feedback**

We sincerely thank all reviewers for their time and effort in evaluating our paper. Your insights and feedback have been invaluable in further strengthening our work.

Below, we address each of your concerns individually. Additionally, we have made corresponding updates to the paper, which are highlighted in blue for clarity and convenience.

---

### Decision · Action_Editor_SfbR · 2025-04-15

**Recommendation:** Accept with minor revision

**Comment:**

This paper identifies a problem with a previously presented unsupervised node embedding algorithm, PCAPass: while concatenation of multi scale embeddings helps avoid over-smoothing, it can lead to rank deficiency, which can negatively impact performance. The paper proposes an alternative approach that avoids this problem.

The reviewers were somewhat lukewarm on the paper due to its potential lack of impact. I agree with the reviewers that the impact may be small, since the field is dominated by SSGNNs, and since PCAPass is not a commonly used approach. However, per TMLR guidelines, this is not sufficient justification to reject the paper. I believe the paper will be of interest to some in the field, and disagree with the idea that there is no value to large-scale graph methods that are not inductive.

I would like to see some expansions of the discussion/analysis, however:
1. Does the concatenation issue identified in PCAPass also impact SSGNNs that use concatenation?
2. How does the (big-O) computational cost compare with PCAPass?
3. For the graph alignment section, it is hard to see whether ACC is performing "well", vs "better than PCAPass". I would like to see a strong baseline here (you don't necessarily need to beat a strong baseline, but it would help to know whether this method would be a good choice for graph alignment).

I am also sympathetic to Reviewer 7mXH's concern about the hyper parameters of the SSGNNs. I would consider performing a lightweight hyper parameter search if computational budget allows, or including published results on the same datasets.

**Audience:**

Yes, this would be of interest to at least some individuals interested in graph embeddings.

**Claims And Evidence:**

The claims are supported by both theoretical and empirical evidence. The authors have expanded their discussion of why there is a problem with the existing PCAPass algorithm, and provided empirical evidence of improved performance over PCAPass and comparable performance to many supervised approaches.

---

> ### Author Response · Authors · 2025-05-09
> **Response to Action Editor**
>
> We would like to sincerely thank the Action Editor for coordinating the review process and for recognizing the contribution of our work.
>
> We have revised the manuscript to address all points raised in your summary:
>
> 1.  **Rank Deficiency in SSGNNs**: In Section 6, we now include a new discussion exploring how the rank deficiency issue may also manifest in self-supervised graph neural networks (SSGNNs). This is supported by additional experiments presented in Appendix G.
>
>  2. **Time Complexity Analysis**: A detailed big-O complexity analysis of ACC, along with a comparison to PCAPass, has been added in Section 4.1. This analysis formally demonstrates the computational benefits of ACC.
>
> 3. **Graph Alignment Baselines**: To contextualize ACC’s graph alignment performance, we compare it to three state-of-the-art alignment methods: S-GWL, CONE, and FUGAL. These results are included in Appendix E to complement, without distracting from, the primary ACC vs. PCAPass comparison in Section 5.1.
>
> 4. **Hyperparameter Tuning for SSGNNs**: In Appendix H, we now include a discussion of hyperparameter tuning challenges in unsupervised settings, along with the results of a lightweight tuning experiment for key hyperparameters (embedding dimensionality and training epochs). This complements our main results and addresses concerns about fairness in the SSGNN comparison.
>
> We hope these additions clarify the scope and implications of our contributions and help underscore ACC’s relevance as an efficient, rank-preserving alternative to existing methods.
> References
>
>     [S-GWL] Xu, H., Luo, D., & Carin, L. (2019). Scalable Gromov-Wasserstein Learning for Graph Partitioning and Matching. NeurIPS. https://openreview.net/forum?id=S1lMD4BgLB
>
>     [CONE] Chen, X., Heimann, M., Vahedian, F., & Koutra, D. (2020). CONE-Align: Consistent Network Alignment with Proximity-Preserving Node Embedding. CIKM. https://doi.org/10.1145/3340531.3412136
>
>     [FUGAL] Bommakanti, A., Vonteri, H. R., Skitsas, K., Ranu, S., Mottin, D., & Karras, P. (2024). FUGAL: Feature-fortified Unrestricted Graph Alignment. NeurIPS. https://openreview.net/forum?id=SdLOs1FR4h